# Cognitive Steering in Deep Neural Networks via Long-Range Modulatory Feedback Connections

**Talia Konkle**
Department of Psychology, Kempner Institute
Center for Brain Sciences
Harvard University
Cambridge, MA 02138
talia_konkle@harvard.edu

**George Alvarez**
Department of Psychology, Kempner Institute
Vision Sciences Laboratory
Harvard University
Cambridge, MA 02138
alvarez@wjh.harvard.edu

## Abstract

Given the rich visual information available in each glance, humans can internally direct their visual attention to enhance goal-relevant information—a capacity often absent in standard vision models. Here we introduce cognitively and biologically-inspired long-range modulatory pathways to enable 'cognitive steering' in vision models. First, we show that models equipped with these feedback pathways naturally show improved image recognition, adversarial robustness, and increased brain alignment, relative to baseline models. Further, these feedback projections from the final layer of the vision backbone provide a meaningful *steering interface*, where goals can be specified as vectors in the output space. We show that there are effective ways to steer the model that dramatically improve recognition of categories in composite images of multiple categories, succeeding where baseline feed-forward models without flexible steering fail. And, our multiplicative modulatory motif prevents rampant hallucination of the top-down goal category, dissociating what the model is looking for, from what it is looking at. Thus, these long-range modulatory pathways enable new behavioral capacities for goal-directed visual encoding, offering a flexible communication interface between cognitive and visual systems.

## 1 Introduction

In any given view there can be multiple kinds of objects present, each with a variety of features and properties. Humans have the capacity to direct their encoding of the visual world based on their internal goals, e.g. when looking for keys, flexible top-down attention mechanisms select and amplify the relevant key-like image statistics [1, 2, 3, 4, 5]. However, while standard deep neural network vision models can accurately classify objects, a key limitation of these visual encoding models is their inability to more deeply integrate with cognitive systems–that is, to encode the visual world in a goal-directed manner [6, 7]. Drawing on insights about top-down attention in both neuroscience and visual cognition, here we design and implement long-range modulatory (LRM) feedback connections that can be added to feed-forward deep neural network models, to enable goal-directed 'cognitive steering' of visual processing.

The idea of adding feedback connections and recurrent processing in deep neural network models is certainly not new (see [8] for review). However, this research typically focuses on characterizing the efficiency benefits of reusing features (e.g. [9, 10]), and does not explore the potential for goal-directed encoding. Other prior research aims more directly at adding goal-directed encoding into deep neural networks, whether using long-range feedback or alternative mechanisms (e.g. [7, 11, 12, 13, 14, 6]). However, these models typically operate over simpler datasets, require customized rather than flexible goal-directed encoding, and/or suffer from strong hallucinations of the goal-directed category

37th Conference on Neural Information Processing Systems (NeurIPS 2023).

arising from strong additive attention signals. Thus, this present work uniquely leverages long-range modulatory feedback to enable goal-directed steering of visual processing.

Here we draw on research in the biological and cognitive sciences, which provides several constraints on how steerability might be implemented in deep neural network models. Anatomically, top-down attention relies on long-range feedback connections, which are extensively found in the neural system [15, 16, 17, 18]. For example, frontal (cognitive) areas project to object-responsive IT cortex, a key endpoint of the ventral visual encoding stream [19]. And within the ventral visual stream, there are extensive feedback connections, with prominently studied pathways from IT cortex to V4, and from V4 to V1. Indeed, recent evidence using optogenetic suppression of feedback in non-human primates demonstrated that goal-directed attention is causally dependent on top-down feedback [20]. Further, cognitive research has shown that feature-based attention obligatorily operates over the full-field (e.g. if you want to attend to faces on the left side of the image, faces on the right side of the image will also be amplified [21, 22, 23, 24, 25]). We leverage these known characteristics of the biological system to guide our implementation of long-range modulatory feedback projections.

We first introduce long-range modulatory feedback models (LRM models), and describe the details of their design, focusing on their distinctive multiplicative modulatory motif (Fig. 1). Next, we probe the default dynamical behavior of these models. Finally, we show that these learned feedback pathways can be used to enable cognitive steering, and we explore which kinds of steering vectors are most effective at recognizing objects in multi-class composite images.

The main contributions of this paper are as follows. (1) Models with long-range modulatory feedback pathways (LRM models) naturally learn more accurate and adversarially-robust representations, relative to a matched baseline model. (2) Feedback projections from the source layer provide a meaningful *steering interface*, and there are effective ways to steer the model which dramatically improve recognition of categories in composite images. (3) The multiplicative modulatory motif prevents rampant hallucinations/false alarms when the goal category is not present. (4) These LRM models learn feature tuning that is naturally more aligned with neurophysiological data; and with active steering, these models qualitatively reproduce effects of top-down category-based attention observed in human brain responses. Broadly, these results demonstrate how long-range modulatory feedback pathways allow for different goal states to make flexible use of fixed visual circuity, supporting dynamic goal-based routing of incoming visual information.

## 2 Models with Long-Range Modulatory (LRM) Feedback Projections

We developed long-range modulatory (LRM) projections, implemented in PyTorch as a wrapper that can augment any PyTorch model with the ability for any source layer to modulate the activity of any other destination layer[1]. We then outfitted standard Alexnet architectures with a variety of different feedback pathways (Sec 2.1). We detail how these networks operate in the next three sections (Fig. 1). Sec 2.2 describes the macro-scale flow of visual information through an LRM model, clarifying *when* incoming feedback signals influence the next processing step of any given layer. Sec 2.3 digs into the details of *how* this feedback signal is computed, and what is learned. Sec 2.4 covers how these models were trained with a two-step loss function.

### 2.1 Architectures

Which layer-to-layer feedback pathways should be included? There is a large space of possible architectures, even in a simple 8-layer Alexnet. If we were to draw on insight from the neuroanatomy of the ventral stream, *all* layers should project back on *all* earlier layers (e.g. [17]). Here, we started more simply, implementing three progressively rich long-range modulatory models–LRM1, LRM2, and LRM3–with an increasing number of feedback pathways. The LRM1 model includes modulatory pathways $(Output \rightarrow Conv4)$ and $(Conv4 \rightarrow Conv1)$, loosely inspired by the known prominent connections from IT to V4 and from V4 to V1. The LRM2 model adds $(Output \rightarrow Conv5)$ and $(Conv5 \rightarrow Conv2)$. The LRM3 model further adds $(FC6 \rightarrow Conv3)$. In all cases, the source of feedback was a relu stage (output of a layer block), and the destination was a conv stage (pre-relu). Critically, the output layer is the source of some of these feedback projections, which will later be important for cognitive steering.

---

[1]Code available here: https://github.com/harvard-visionlab/lrm-steering

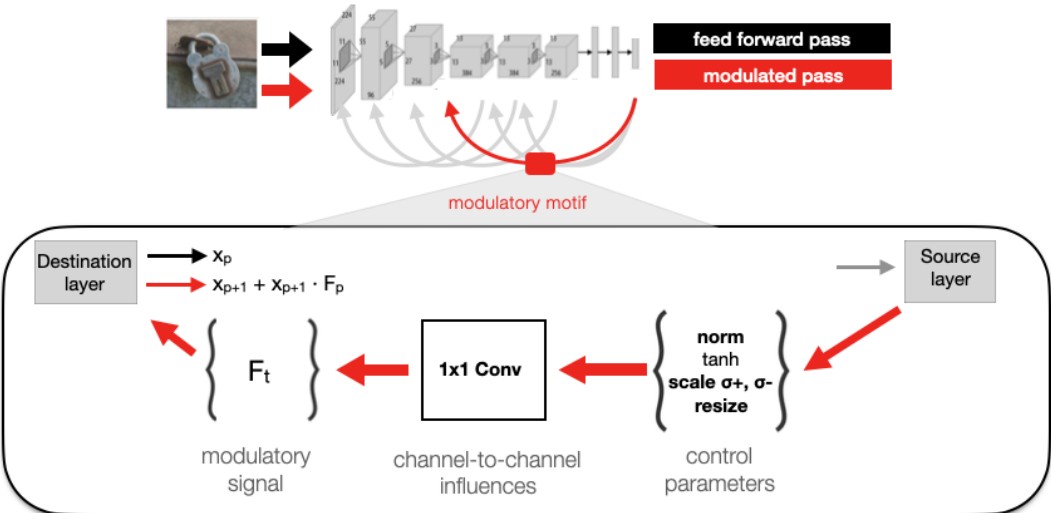

Figure 1: Schematic of long-range modulatory (LRM) network architecture. The image is processed in the initial feed-forward pass, and then source layers modulate destination layers in the modulated pass. Learnable parameters of the modulatory motif in bold.

## 2.2 Long-range modulatory pathway dynamics

In the standard feed-forward operation of the base vision network, the flow of the information proceeds sequentially from the input layer to output layer. In a model with LRM projections, the flow of information through the network can be best understood as *consecutive feed-forward passes*.

In the **initial feed-forward pass** ($pass = 0$), the output of layer $L$ is:

$$x_p^{out} = L(x_p^{in}) \tag{1}$$

After this pass, layers that are the source of feedback pathways store their output activations $x_p^{out}$, to influence destination layers on the next forward pass. During the next feed-forward pass of the image—also referred to here as the **modulated pass**—each layer first computes its typical feed-forward output. Then, the layer incorporates feedback influences coming from any sources in the prior pass ($F_p$), with the following multiplicative effect:

$$x_{p+1}^{out} = L(x_{p+1}^{in}) + L(x_{p+1}^{in}) \cdot F_p \tag{2}$$

With this motif, the source layers provide a *multiplicative, modulatory influence* on the destination layer's feed-forward signal. In the absence of feedback drive, the feed-forward signal is unchanged. And, in the absence of input, the feedback pathways do not drive activations through the network.

## 2.3 Modulatatory motif

How are the source layer activations transformed to modulate the destination layers on the next pass ($F_p$)? The modulatory motif we implemented is motivated by the hypothesis that there are meaningful relationships between channels in the source layer and the destination layer (e.g. [26]). The motif is schematized in Fig. 1 and detailed mathematically in Appendix A.1, and summarized below.

First, source layer activations undergo a series of normalizing and scaling transformations to control and align their modulatory influence with the destination layer. Briefly: (1) A channel normalization operation serves to emphasize some features, and de-emphasize others, over a given image, with learnable normalization parameters. (2) To keep the multiplicative modulation values in a controlled range, normalized activations are next passed through a $tanh$ function. We include two learnable scalars $\sigma+$ and $\sigma-$, to enable the model to learn how to best scale the positive and negative modulatory

influences, respectively. (3) Next, the size of the source activation map is aligned to the size of the destination map. (If the source layer is fully-connected and the destination layer is a convolutional layer, the channel activations are spatially broadcast across the full activation map. If both the source and destination layers are convolutional, the source activation map is spatially up-sampled to align with the size of the destination activation map.)

Next, the key transformations carried out in these LRM pathways are the ***channel-to-channel influences*** that are learned between the source features and the destination features. This stage was implemented through a 1x1 convolutional layer, and outputs a feedback map $F$ with the same size as the destination layer activation map. To give an intuition, these channel-to-channel weighting parameters enable long-range projections to learn that some feature channels in the source layer should amplify particular feature channels in the earlier stage, and suppress other feature channels, to help correctly classify the image on the next pass through the network.

### 2.4 Loss Function and Training

All LRM models were trained on the ImageNet dataset [27, 28] on the task of image classification. The total loss was calculated as the average of the cross-entropy $CE$ loss on the predictions of the initial feed-forward pass $y'_{p=0}$ and one modulated pass $y'_{p=1}$, given ground truth labels $y$:

$$L = \frac{1}{2}(CE(y, y'_{p=0}) + CE(y, y'_{p=1}))$$

(3)

The motivation for this two-term loss function is that we wanted the base encoder to learn an effective set of visual features for pure feed-forward operation (as is true of the biological system). And, we wanted to ensure that learned feedback modulation routing could further reinforce the feed-forward operation for successful object classification. Note that in this work, only two passes were ever trained (one initial feed-forward pass, and one modulated pass).

Models were trained using the Fast-Forward Computer Vision (FFCV) library [29]. We used a one-cycle policy [30] over 100 epochs, with 1e-4 * peak-learning-rate as the initial rate, 0.10 as a the peak at epoch 15, approaching 0 as the final rate. We used a batch size of 2048, a weight decay of 5e-5, and momentum of 0.9, with the SGD optimizer. Models were trained on an internal computer cluster with 48 cores, 500GB of system RAM, and 4 A100 GPUs with 40GB RAM each. LRM-Alexnet models train in less than 8 hours on this hardware.

## 3   Model Performance During Default Operation

First, we characterize the performance of these LRM models during their default modulatiory operation. Note there is no goal-based cognitive steering yet–the model simply is carrying out its default modulatory feedback processes at inference time.

### 3.1   LRM models have increased accuracy and robustness

How well do the LRM models classify images on their feed-forward pass and modulated passes, and how does that compare to a baseline model without LRM pathways? Figure 2A shows top-1 classification accuracy, plotted for all three LRM models, as a function of modulatory passes tested at inference. Note that all LRM models were trained on only the initial pass (pass 0) and first modulatory pass (pass 1), but we allowed further modulatory passes at inference to explore these effects.

All LRM models achieved higher accuracy than a comparison Alexnet backbone architecture, trained with the same recipe, but without any LRM pathways (dashed gray line). In addition, in all models, the modulated passes had higher accuracy than the initial feed-forward pass, with accuracy plateauing after the second modulatory pass. An implication of this result is that some images are initially incorrectly classified, but with default feedback modulation on the next pass, the image maximally activates an alternate, correct class. We show an example of such a case, using the AblationCam to observe the shifting evidence between the incorrect and correct classification, in Fig. 3).

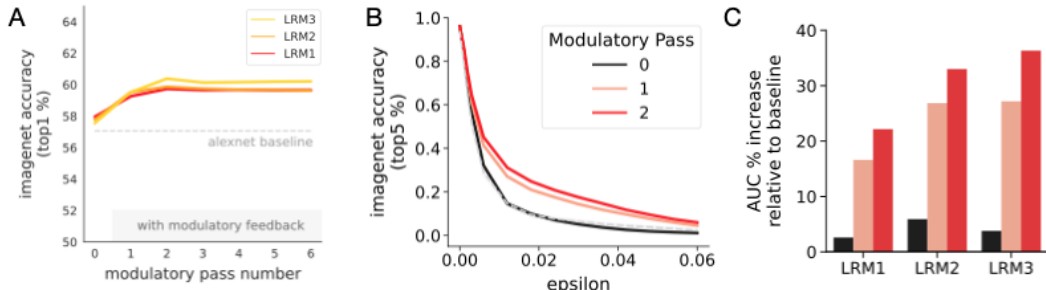

Figure 2: LRM model default behavior. (A) Imagenet Accuracy plotted for the three LRM models, as a function of initial and increasing modulatory passes. (B) Adversarial robustness is plotted for the LRM3 model, as a function of epsilon. (C) Percent improvement in adversarial robustness, relative to the standard Alexnet backbone, is plotted for all three LRM models. Dashed gray lines in (A), and (B) indicate performance of a matched Alexnet baseline, with no LRM pathways.

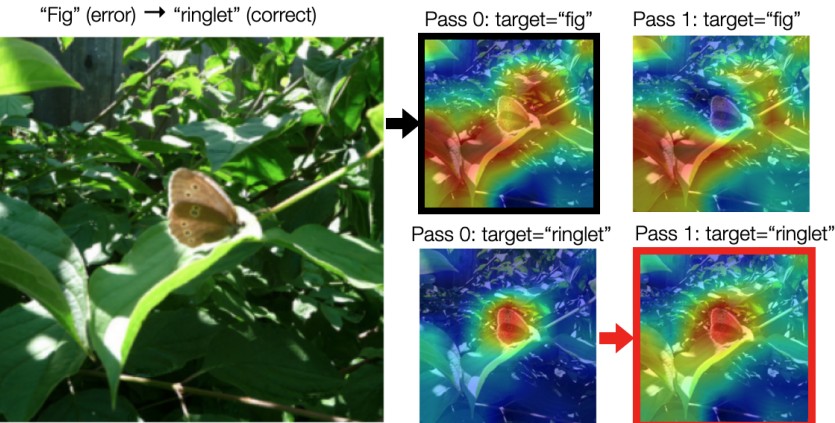

Figure 3: Example of modulatory-pass improvement. Image is initially incorrectly classified as a "fig", and then correctly classified as "ringlet" (moth). AblationCam [31] shows the evidence with respect to these two output target units. Initially the depicted leaves dominate the classification, with some evidence for "ringlet" focused on the depicted moth. During the modulated pass, however, the "fig" output unit no longer depends on the depicted moth in the image, resulting in the "ringlet" becoming the maximally activated output unit.

Note that LRM models have more trainable parameters, which could underlie this difference in accuracy. However, even the 0th (feed-forward) pass of the LRM models was slightly but consistently higher than the baseline Alexnet, indicating that the LRM models actually have learned *better* feed-forward features than the matched architecture model without feedback projections.

In addition, we also found that LRMs have increased robustness to adversarial attacks, using Fast Gradient Sign Attack [FGSM 32]. Fig. 2.B shows the LRM3 model robustness, plotted for the feedforward pass and the first two modulatory passes. Fig. 2.C summarizes the robustness effect for all models (summed AUC over the epsilon range), plotted as a percentage increase in robustness, relative to the control Alexnet model. These results demonstrate that the addition of long-range modulatory pathways naturally confer adversarial robustness in their default modulatory operation.

Why is the modulatory pass so effective? We hypothesize that learned feedback connections are reinforcing the implicit routes that images take through the computational hierarchy (e.g. [33, 26, 34]). If images are ambiguous between multiple classes at the later source stage, normalization-based competition may shift the evidence about which class is present. Learned feedback signals then amplify relevant features and suppresses irrelevant features in the earlier stages, to steer the input along different routes in the modulated pass. An alternative possibility, or framing, is that top-down connections provide a holistic view of the scene, across both spatial and channel dimensions, and

serve to edit early activations to be most consistent with the current, holistic high-level representation (see [35] for related ideas on hierarchical Bayesian inference in visual cortex).

## 3.2 LRM models have higher Brain-Scores

To examine whether these brain-inspired feedback projections also yield better representational alignment between the model and the brain data, the LRM3 model was submitted to the Brain-Score benchmark [36], and results are reported in Table 1. We find that the LRM3 model shows much improved predictivity of the later stage of the ventral visual cortex (IT), dramatically better than PyTorch default Alexnet (e.g. model rank 145 → 35 across submitted models, at the time of this analysis). More generally, the LRM3 model with default feedback operation shows a better fit to three of the four visual brain regions, both in its feedforward pass and its modulated pass, compared to the PyTorch default Alexnet model.

| Brain Area | Baseline Alexnet | LRM3 (pass 0) | LRM3 (pass1) | Δ (from baseline) | rank change |
|---|---|---|---|---|---|
| IT | r = 0.358 | r = 0.393 | r = 0.400 | **+0.042** | #145 → **#35** |
| V4 | r = 0.443 | r = 0.454 | r = 0.467 | **+0.024** | #153 → **#97** |
| V2 | r = 0.353 | r = 0.341 | r = 0.333 | -0.020 | #13 → #48 |
| V1 | r = 0.507 | r = 0.492 | r = 0.531 | **+0.024** | #68 → **#32** |

Table 1: Brain-Score results for the Baseline Alexnet model and LRM3 model. The r-value indicates the average single-unit neuron predictivity scores, reported for different visual areas along the ventral visual stream hierarchy.

# 4 Model Performance with Cognitive Steering

We next show that these learned feedback pathways can be co-opted for goal-directed cognitive steering. The underlying idea is intuitive: the output layer is the source of feedback projections that have already learned how to effectively modulate feed-forward visual processing. And, the output latent space is interpretable, and thus can be thought of as cognitively-accessible *steering interface*. In this 1000-d space, different vectors $v$ can be meaningfully interpreted as steering towards (or away) from different categories (Fig. 4). During goal-directed *cognitive steering*, we simply intercept the default feedback sent from the output source projections, and instead, inject a different output activation pattern to drive the modulatory feedback pathways.

Note that a premise of cognitive steering is that you have a goal or target in mind. As an example, the ongoing cognitive task may next require the agent to look for a key. In this case, the cognitive system may retrieve the "prototypical" output vector for the "key" class from memory, which could be used as a top-down steering vector, and be injected into the feedback projections. We predict that cognitive steering should allow the visual encoder to amplify any responses for key-like features that are present, enabling better detection. And, because the feedback is purely modulatory, this motif should prevent the model from hallucinating keys everywhere.

## 4.1 Composite Image Recognition Challenge

To test these ideas systematically, we used a composite image recognition task, inspired by cognitive science research testing effects of top-down feature based visual attention [37, 5]). Specifically, composite images were created, consisting of two images of different classes either presented side-by-side, or directly overlaid on top of each other and averaged. The logic here is that these composite images will be hard for models to classify in typical feed-forward operation, but either class will be accurately recognized with corresponding cognitive steering. And, conversely, if we cognitively steer the model toward a category that is not present in the composite images, the model should not hallucinate that the goal-category is present (i.e. false alarm).

First, we created a controlled composite image test, using the 'Imagenette' subset of 10 categories to create a set of image triplets (3,910 image triplets). Each triplet was created by randomly drawing 3 images from 3 distinct categories (without replacement). The first two of the images were used to form a composite image with two "target-present" categories, by either placing them side-by-side

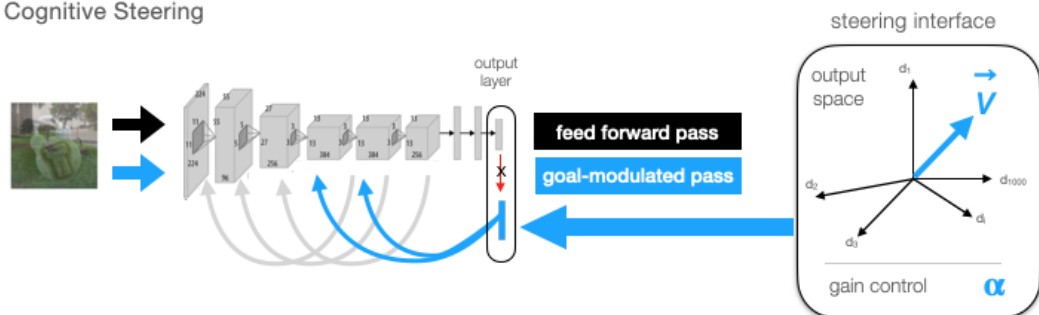

Figure 4: Cognitive Steering. Feedback pathways from the output layers are used for goal-directed influence over the modulated pass. The output latent space serves as a cognitive-steering interface, where any directional vector ($v$) serves as the guidance signal. The gain of the modulatory signal can be controlled with $\alpha$-scaling.

or overlaid (Fig. 5.A). The third image from the triplet was kept aside to be used a "target-absent" category to test for steering-induced hallucinations, i.e. whether steering towards a non-present category leads the model to report a category that was not present in the image.

The baseline Alexnet model is very poor at identifying either of the two classes of the composite images correctly: side-by-side composite recognition = 23.6%[2]; overlay composite recognition = 1.0% (top-1 accuracy). It can, however, very accurately classify these images when presented individually: top-1 accuracy = 77.0%. Thus, these composite images provide a good challenge for testing whether cognitive steering can improve classification, and to what degree.

### 4.2 Cognitive Steering Variations

How should a cognitive system effectively steer the models–that is, what vector should be used as the source of the feedback to the vision backbone, to enable the most accurate classification of a category in these composite images? We designed and compared a variety of steering signals, reporting only a subset here for clarity (see also Appendix A.5). Here we focus on several forms of target-based steering, which aim to steer the model towards a representation of the target of interest, with different possible levels of granularity:

- *One-hot steering*, using a 1000-d vector with a 1 on the target category unit.
- *Instance-steering*, using the 1000-d activation profile of the output layer when the network processes that exact target image, in isolation.
- *Category-prototype steering*, using a 1000-d vector of the average output activation, over all exemplars of the target category.
- *Language-based category steering*, leveraging a diffusion-prior model [38] to map between CLIP-text outputs and the LRM visual embedding space (see Appendix A.3), providing a text-based prediction of the output activation expected for the target category from the vision model.

Each of these steering signals operationalizes a different high-level query for how the cognitive system can interact with the visual system to guide visual encoding (e.g. "look for this specific image" or "look for anything from category X").

We also explored whether we could vary steering strength. LRM models have learned $\sigma^+$ and $\sigma^-$ scaling parameters that control the magnitude of modulation. We probed whether scaling these values by $\alpha$ has systematic effects on the strength of the cognitive steering influence ($\alpha \cdot \sigma^+, \alpha \cdot \sigma^-$, for $\alpha = 1, 2, 3, 4$, Fig. 5.B). In human vision, this factor is loosely equivalent to generalized cognitive effort, and our capacity to marshall increasing resources to amplify top-down effects [39].

---

[2]The side-by-side images have 2x the horizontal pixel resolution. Our model uses PyTorch's standard, built-in Adaptive Average Pooling operation at the final convolution stage to adjust the size of the representation to match the size expected by the fully-connected layers.

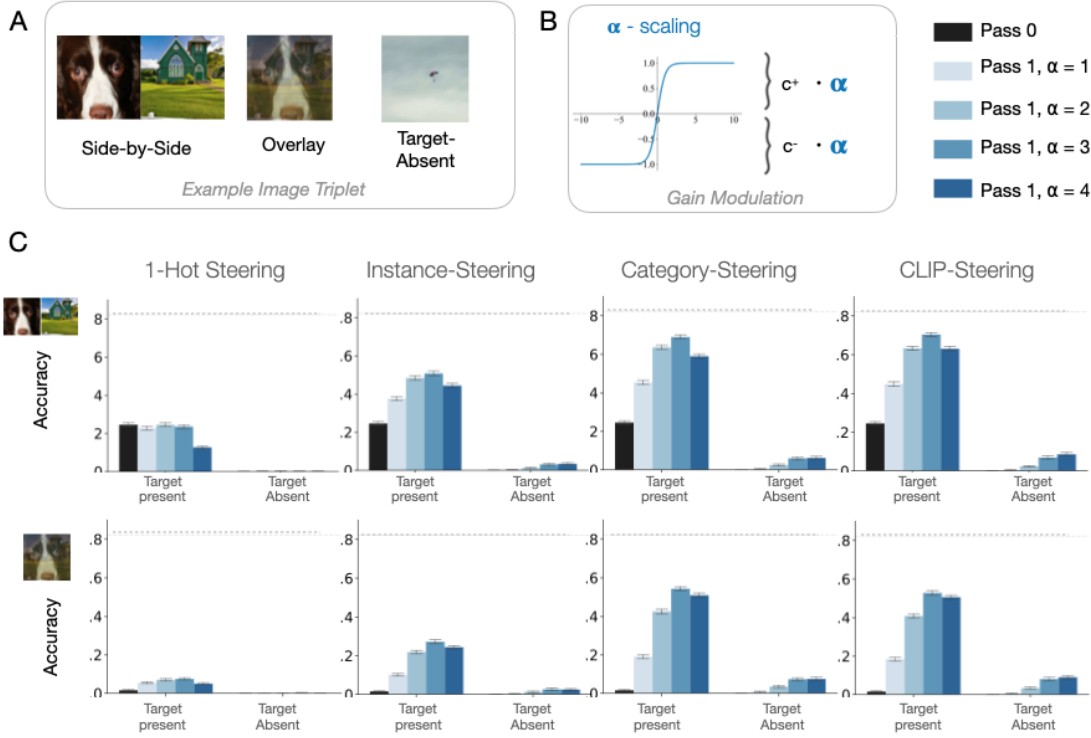

Figure 5: Cognitive Steering Tests. (A) Example triplet from the controlled composite tests. (B). Schematic of gain modulation mechanism. (C). Results of LRM 3 architecture are shown, for both side-by-side and overlay composites tests. Subplots show different cognitive steering signal variations. Target-present conditions indicate percent of times the target category was predicted, when steering towards that target. Target-absent condition indicate the percent of times that the target-absent category was predicted, when steering towards the target absent category (false alarm). Error bars indicate the 95% bootstrapped confidence interval across composite images. Horizontal, dashed gray lines indicate the accuracy of classifying the individual target images alone, computed for the LRM3 model for its default modulated pass.

### 4.3  Cognitive Steering enables flexible recognition in composite images

The results of the LRM3 model with various forms of cognitive steering are shown in Fig. 5.C. We see that without any steering (in the feed-forward pass 0, black bars), the model is unable to accurately identify either category from these composite images: side-by-side=24.5% , overlay=5.1%. We found that cognitive steering was generally able to improve recognition of either category, in some cases, quite dramatically. For example, category-prototype steering (with $\alpha$-scaling=3) showed side-by-side recognition = 68.9% and overlay recognition = 51.8%. Language-based steering (also with $\alpha$-scaling=3) yielded side-by-side recognition = 70.4%, overlay recognition = 50.5%. To contextualize these effects, the level of accuracy obtained by this model using its default modulated processing pass over the individual images is 82.2% (dashed gray lines Fig. 5.C). Finally, gain control was also particularly effective for category-prototype and CLIP-steering, with increasing $\alpha$-scaling following a u-shape curve (for $\alpha > 4$, accuracy continues to decrease). Comparable results were observed for the LRM2 model, and much weaker results for the LRM1 model (see Table A.4). Appendix Fig. 7 provides an additional plot highlighting the effect cognitive steering on the logits.

Broadly, the pattern of results show that category prototype steering and CLIP steering were the most effective. This pattern is perhaps surprising, revealing that a more graded, probabilistic steering target is more effective than steering based on the objective of the model or through the specific representation of the exact image. Taken together, these experiments demonstrate that these LRM feedback pathways enable effective, parameterizable cognitive steering, yielding dramatic improvements in recognizing the target categories present in composite images.

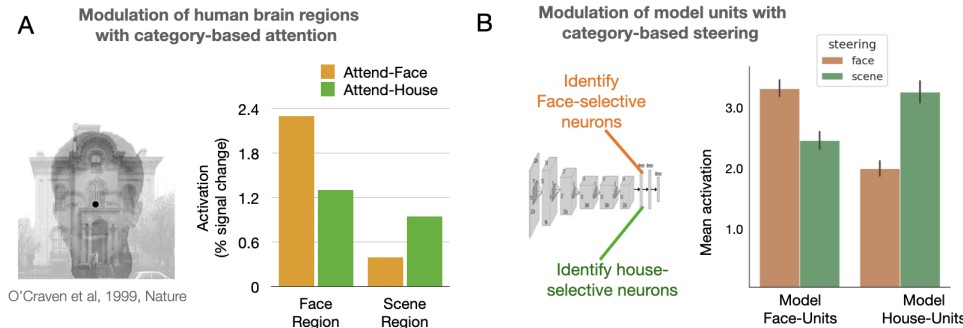

Figure 6: (A) Modulation in face- and scene-selective visual brain regions, during top-down attention of face-scene composite images [41]. (B) Parallel results in LRM networks, where face-tuned and scene-tuned units from a late stage layer of the model show different levels of activation to a composite image, when steering by face- and house-prototypes.

## 4.4 Multiplicative modulation prevents rampant hallucinations

We next tested whether cognitive steering towards a non-depicted category would lead to increased false recognition. For example, given a composite of *English Setter* and *Church*, what if the model is steered to "look for" *Parachute*? For these tests, we used the representation of the target-absent image of each triplet for cognitive steering. The results show that the likelihood of classifying the composite image towards the absent, non-target category was negligible, for all steering types (Fig. 5).

In follow-up analyses, we directly tested the claim that the multiplicative feedback is critical for preventing excessive false alarms to the steered category. First, we implemented matched LRM models with additive feedback projections. Our (standard) LRM-*multiplicative* motif is $L(x_{p+1}) + L(x_{p+1}) \cdot F_p$; we trained additional LRM-*additive* models that influence destination layers additively: $L(x_{p+1}) + F_p$ (Appendix A.7). Second, we also considered a CORnet model, a brain-inspired recurrent neural network model with additive feedback of a layer, onto itself, over time [40], and invented a different way to inject a steering signal into its recurrent processes (Appendix A.8). We found that steering LRM-*additive* models and CORnet models did improve detection of the goal category, but also lead to increased false alarms. These comparisons help clarify the benefits of multiplicative over additive feedback modulation: while both motifs can boost target-aligned features, only the multiplicative motif maintains low false alarms, even when increasing attentional gain.

## 4.5 Cognitive Steering recapitulates neural signatures of category-based attention

In the biological vision sciences, extensive research has shown that goal-based attentional states modulate activity along the entire visual processing hierarchy (e.g. evidenced across methods, including single-unit electrophysiology [5, 42], functional magnetic resonance imaging [41, 43], and magnetoencephalography [21, 44]; for review see [45, 46, 47, 48]). For example, seminal empirical work showed that when participants viewed composite face-house images, their face-selective brain regions were more active with goal-directed attention to faces, while scene-selective regions were more active with goal-directed attention attention to scenes, even though the actual composite visual input was always the same (Fig. 6.A; [41]).

Our LRM models recapitulate this well-known signature of top-down category-based attention, with systematic modulation of face-selective and scene-selective units based on the steering target (Fig. 6.B; see Appendix Sec.A.6). To our knowledge, this is the first mechanistic model, operating over rich naturalistic images, to account for these empirical neural signatures (for the most related earlier work see [7, 11]). Further, to our knowledge, current feed-forward/recurrent vision models do not (yet) have mechanisms to support top down-goal based modulation of early visual processing stages.

## 5 Limitations

The three LRM models tested here reflect only a small part of the space of possible long-range modulatory architectures. Our results show increasing benefits with more long-range modulatory

pathways, leaving open the search for the optimal macro-scale architecture design. Additionally, it is an open question whether similar improvements in accuracy and adversarial robustness would emerge if LRM pathways were added to standard ResNets [49] or state-of-the art Convnexts [50] or Vision Transformers [51]. We note, however that even if improved accuracy and robustness are not observed in larger models during default operation, adding LRM feedback pathways would still enable new cognitive steering capacities, and is thus an important avenue for future work.

In this work, we have not done an analysis of the structure in the learned channel-to-channel weights, nor characterized how these modulations are changing the representational geometry of each layer, nor provided mechanistic clarity on why these feedback connections are effective. And, we have not explored the impact of the learnable normalization components relative to the learnable channel-to-channel weights on the effects reported here, nor considered alternate motifs of more spatially localized (rather than globally broadcast) modulatory influences. Given the behavioral improvements found in these LRM models, deeper exploration into these inter-workings and variations would be valuable next steps.

Finally, while we used category-supervised training here, we think an important next step where the capacities of these LRM models would particularly shine is in a self-supervised setting (e.g. [52, 53, 54]). In particular, both the category prototype steering signals and the language-based signals are completely agnostic to the content of the output space, and can operate equally well in self-supervised domains. Thus, this work sets the stage for cognitive queries to help guide self-supervised visual encoding of the world, enabling new featural distinctions to be emphasized between the feed-forward and modulated pass, related to the goals of the cognitive system.

## 6 Related Research

Recurrent feedback connections have been integrated in deep neural networks in prior research, often drawing inspiration from the extensive back-projections present in the visual system [e.g. 55, 56, 10, 57, 58, 9, 59, 12, 60, 6, 61]. For example, [62] introduced feedback connections to convolutional networks, with a loss function for each time step (as we did here), focusing on the benefits of iterative emergence of categorical information over many successive steps. [7] explored an external cognitive bias implemented through recurrent feedback, to guide spatial selection mechanisms to recognize overlapping digits. [11] used an extensive procedure to search through a million-dimensional parameter space of feedback connection motifs, yielding models that show a similar capacity to correct mis-classification errors with recurrent steps [see also 9].

Our model introduces a relatively simple multiplicative feedback modulation, contrasting with modeling approaches in which feedback activation signals additively influence earlier layer activation [e.g. 10, 57], or more structured approaches which process top-down signals through separate parallel architectural branches [e.g. 63, 58, 64]. We also designed our model to operate with one modulatory pass, enabling rapid goal-directed steering with separate gain modulation–this departs from traditional recurrent neural networks which typically focus on the benefits of recurrance over longer roll-outs [e.g. 62, 56]. Our work focuses on how feedback connections can be used for top-down goal-directed visual encoding–this has parallels to work focusing on top-down regulation, to aid in recognition despite partial occlusion or clutter, or in object localization [e.g. 59, 55, 63, 12, 65, 66]. Top-down feedback networks have also been used to connect with theories of predictive coding, typically in support of more temporally extended processing of bottom-up visual information through time [e.g. 67, 68, 60, 69, 58], whereas here we focus on their use for external guidance signals to influence feed-forward visual processing.

## 7 Conclusions

Here we introduce cognitively- and biologically-inspired long-range modulatory pathways. We show that these learned modulatory feedback routes create fixed circuitry that can be leveraged for more flexible, goal-based modulation of visual input processing, enabling "cognitive steering" of visual processing. Broadly, we suggest that these architectural pathways provide novel targets for the introduction of top-down steering, offering new possibilities for integrative systems (e.g. multi-modal vision-language alignment; RL agents with goal-directed visual encoding) to enable more flexible communication between visual and cognitive components of the models.

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

# A  Appendix

## A.1  Modulatory Motif Detail

Here is a more detailed description of how source activations modulate the destination layer activations. First, a channel normalization operation serves to emphasize some source features, and de-emphasize others, over a given image. We include learnable $u_c$ and $\sigma_c$ parameters across all channels $c$ in the source layer. For a source layer activation in channel $c$, the normalization function $\mathcal{N}$ was defined as:

$$\mathcal{N}(a_c; u_c, \sigma_c) = \frac{a_c - u_c}{\sigma_c}$$

For convolutional layers that operate jointly over channels and spatial locations $(x, y)$, the normalization function was extended to:

$$\mathcal{N}(a_{c,x,y}; u_{c,x,y}, \sigma_{c,x,y}) = \frac{a_{c,x,y} - u_{c,x,y}}{\sigma_{c,x,y}}$$

After normalization, the activations $a'$ are passed through the $\tanh$ function, with learned scaling parameters. Function $\mathcal{S}$ scales the output of the $\tanh$ function based on the sign of the normalized activation:

$$\mathcal{S}(a'; \sigma^+, \sigma^-) = \left\{ \begin{array}{ll} \sigma^+ \cdot \tanh(a') & if\, a' \geq 0 \\ \sigma^- \cdot \tanh(a') & if\, a' < 0 \end{array} \right.$$

where $\sigma^+$ is the learnable scalar for positive activation scaling, and $\sigma^-$ is the learnable scalar for negative activation scaling. Next, the size of source activations $a''$ is aligned to the size of the destination map $D_{w,h}$. If the source layer is fully-connected and the destination layer is a convolutional layer, the source channels' activations $S_c$ are spatially broadcast to the size of the of the destination map activation map.

$$Resize(a''_c) = a''_c \times \mathbf{1}_{D_{w,h}}$$

where $a''_c$ is the activation of each source channel $c$ and $\mathbf{1}_{D_{w,h}}$ is a matrix of ones with dimensions $D_{w,h}$. If both the source and destination layers are convolutional, the source activation map is upsampled to be the size of the destination activation map:

$$\mathcal{U}(a''_{c,x,y}; S_{w,h} \rightarrow D_{w,h}) = Upsample(a''_{c,x,y}, S_{w,h}, D_{w,h})$$

where $a''_{c,x,y}$ represents the activation of the source channel $c$ at spatial location $(x, y)$ and the function $Upsample$ up-samples the source activation map from size $S_{w,h}$ to size $D_{w,h}$. Following the normalization and scaling operations, the key transformation in the LRM pathways is the learned channel-to-channel influences. This is implemented using a 1x1 convolutional layer:

$$f_{c',x,y} = \sum_c W_{c',c} \cdot a''_{c,x,y}$$

where $W_{c',c}$ are the learnable weights of the 1x1 convolution, $a''_{c,x,y}$ is the source activation, and $f_{c',x,y}$ is the resulting activation that will be use to modulate the destination layer's next pass.

### A.2 Impact of steering on logits

Fig. 7 provides a more detailed depiction of the cognitive steering signal effects, focusing the goal-directed pass of the LRM3 model (category-prototype steering, $\alpha$-scaling=3). The output activation unit of target 1, target 2, and the target-absent category are shown. When steering toward the category prototype of the either target, there is substantial increased activation along that output unit dimension. Steering towards the absent, non-target category does not have the same consequence (a benefit of the purely multiplicative modulatory interaction).

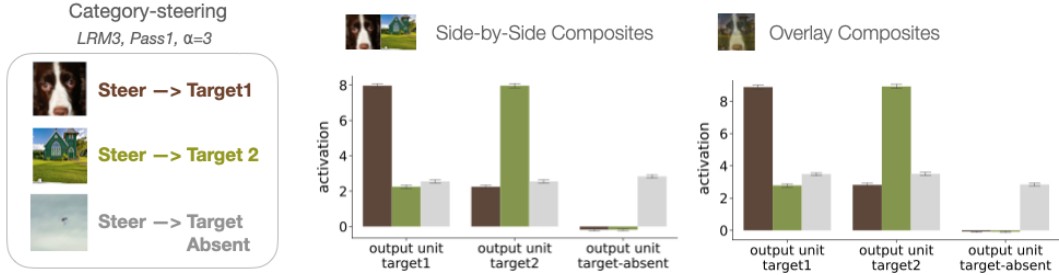

Figure 7: Impact of steering on output unit activation levels (1000 units corresponding to each image category). Average activation is plotted for the output unit corresponding to the category of each triplet image, including the two that were presented (target1, target2) and the image that was absent (absent-image), for both overlay and side-by-side composite images. Error bars reflect standard deviation over images. Data reflect activations from the LRM3 model, after the first modulated pass with category-prototype steering and gain modulation of $\alpha = 3$.

### A.3 Cognitive Steering with Clip Text Embeddings

One potential application of LRM networks is to enable top-down steering of visual encoding via natural-language queries — e.g., steering the visual encoder by asking "Is there an English Springer present?" — amplifying any visual responses aligned with the query throughout the visual encoder. Ideally this language-based steering would be implemented via an integrated vision+language multimodal model, such as CLIP [70], using an LRM network as the vision encoder. We haven't yet trained a multi-modal LRM network, but as a proof of concept for this application, we generated steering vectors from category labels using an intermediate model that was trained to map CLIP text outputs and LRMNet vision outputs. Our approach was modeled after the "diffusion prior" of the Dalle2 architecture [38] using the lucidrains PyTorch implementation (`https://github.com/lucidrains/DALLE2-pytorch`), modified to allow mapping from N-D to M-D feature spaces, since our vision models had a different output dimension than the CLIP text encoders. The goal was to provide a category label, e.g., "English springer", to a CLIP text encoder, and to then map the CLIP text output to the expected LRM network output for an image from that category, specifically to the mean vision output for that image category. This predicted embedding could then be used as a steering vector to exert top-down control over visual processing within the LRM network (see "clip-text prototypes", Sections 4.2 and 4.3).

#### A.3.1 Image Embeddings

We first extracted the output vector for each image in the ImageNet training set, then averaged across images within each category, yielding a 1000x1000 tensor of "visual category prototypes" (1000 categories x 1000 output units). We also computed the prototypes for the ImageNet validation set. These visual category prototypes were computed separately for each LRM network.

#### A.3.2 Text Embeddings

We used the "ViT-B/16" text encoder from the official open source CLIP models `https://github.com/openai/CLIP`. To generate clip-text embeddings for each ImageNet category, we used the zero-shot imagenet labels and prompts from the original CLIP paper [70], as described here `https://github.com/openai/CLIP/blob/main/notebooks/Prompt_Engineering_for_ImageNet.ipynb`. Each ImageNet target label was converted to the ImageNet

classname (e.g., label=0, classname = "tench"), and each classname was included within seven zero-shot prompts ("a photo of the large {}.", "a {} in a video game.", "art of the {}.", "a photo of the small {}.", "itap of a {}.", "a bad photo of the {}.", "a origami {}."). These text prompts were then passed through the text encoder and the outputs were averaged across prompts, yielding a 512-D text embedding, and 77x512 text encoding (embedding per token, max sequence length 77) for each ImageNet category. These clip-text representations could then be used to predict the image embeddings of an LRMNetwork.

### A.3.3 Text-to-Image Diffusion Model

Following [38], we trained a diffusion model to generate image vectors conditioned on text embeddings and encodings. We used the lucidrains DiffusionPriorNetwork and DiffusionPrior modules: prior_network=DiffusionPriorNetwork(dim=1000, depth=6, dim_head=64, heads=8), and DiffusionPrior(net=prior_network, image_embed_dim=1000, timesteps=100, cond_drop_prob=.20, condition_on_text_encodings=True, image_embed_scale=True). In order to align the text representations (512-D) with the image representations (1000-D), we used a linear projection layer (512 input channels, 1000 output channels), and then passed these 1000-D text representations into the Diffusion Prior to condition the diffusion process. The output of the Diffusion Prior is a 1000-D vector, which serves as the predicted visual category prototype for the class specified by the text label. Each batch was composed of the full set of 1000 text-embeddings and 1000 image embeddings, and the models were trained for 20000 iterations using the AdamW optimizer (lr=.0001, betas=(0.9, 0.999), weight_decay=0.01).

Finally, after a DiffusionNetwork was trained, we could "sample" text-conditioned image embeddings from the DiffusionPrior for each of the 1000 ImageNet labels. We aggregated across 100 samples per category to generate "clip-text prototypes", which we treat as the final mapping between ImageNet text labels and the visual encoder outputs.

### A.3.4 Text-to-Image Embedding Quality

To measure the accuracy of the text-to-vision alignment, we computed the cosine similarity between the 1000 clip-text prototypes, and the visual category prototypes for the training set and the validation set. As shown in Table A1, the clip-text prototypes for all models showed high cosine similarity scores on both the training set (>= .938%) and validation set (>= .88%). For comparison, the cosine similarity between training and validation set vision prototypes was .94 for all LRM models.

|  | LRM1 | LRM2 | LRM3 |
|---|---|---|---|
| **Training Set** | 0.950 | 0.944 | 0.938 |
| **Validation Set** | 0.895 | 0.889 | 0.884 |

Table 2: Cosine Similarity between Clip-Text Prototypes and Visual Category Prototypes for each LRM network

### A.4 Composite Test Accuracy, All Models

| Composite Test | Model | Pass 0 | 1-Hot | Instance | Category | Clip |
|---|---|---|---|---|---|---|
| **Side-by-Side** | LRM1 | 0.246 | 0.143 | 0.275 | 0.395 | 0.400 |
|  | LRM2 | 0.236 | 0.109 | 0.382 | **0.743** | 0.718 |
|  | LRM3 | 0.245 | 0.235 | 0.507 | 0.689 | 0.704 |
| **Overlap** | LRM1 | 0.016 | 0.024 | 0.139 | 0.250 | 0.236 |
|  | LRM2 | 0.014 | 0.051 | 0.234 | **0.674** | 0.635 |
|  | LRM3 | 0.016 | 0.075 | 0.272 | 0.543 | 0.527 |

Table 3: Composite test accuracy vs. steering type for all models with alpha modulation 3.

## A.5 Additional Cognitive Steering Variations

We also tested discriminative steering signals that take into account both the targets, in order to amplify one and suppress the other ($v_{target1} - v_{target2}$). These steering vectors effectively query the system to "look for X's as distinct from Y's". To our surprise, we found that discriminative steering did not have a dramatic impact on composite recognition accuracy, relative to target-based steering (Fig 8).

One interpretation of this result is that the modulatory circuitry may have already learned to amplify the target category relative to all other categories, so the additional suppressive steering for the distractor category is not effective. Another possibility is that the particular images used in this side-by-side test are by design relatively distinct, and the discriminative steering signals may be more effective if the target and distractor categories are more similar (e.g kit fox vs arctic fox). Future work will be required to clarify the mechanisms and theory around the most effective cognitive steering signals.

Finally, we also explored whether extra modulatory passes would amplify any steering effects. However, we found that across models and all levels of $\alpha$-scaling and all cognitive steering variations, additional passes had negligible impact beyond the first modulatory pass. This feature of our LRM models further dissociates them from traditional recurrent neural networks, which have the assumed characteristic of running over multiple timesteps times. In contrast, our model has a single parameter ($\alpha$) that controls the gain on the modulation, operating on a single modulated pass.

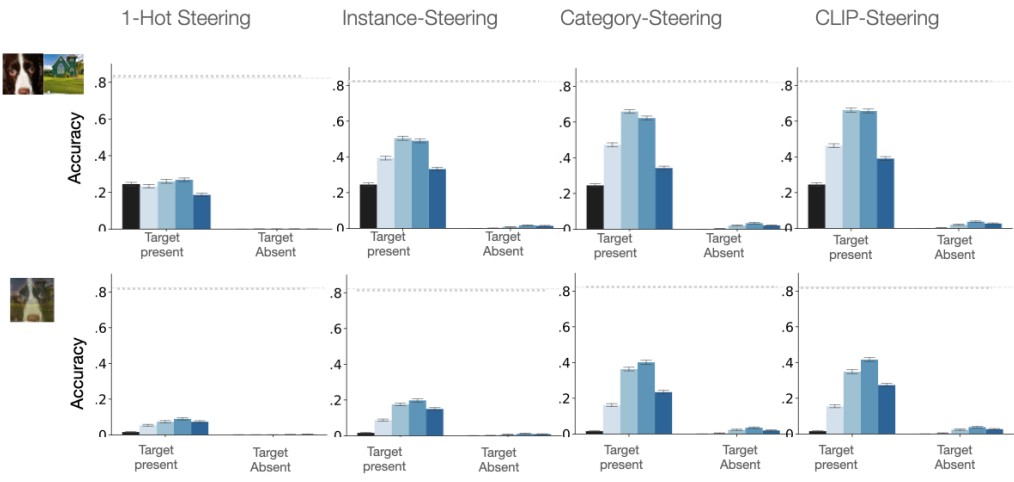

Figure 8: Results of the LRM3 architecture are shown, for both side-by-side and overlay composites tests. Subplots show different *discriminative* cognitive steering signal variations. Target-present conditions indicate percent of times the target category was predicted, when steering towards that target. Target-absent condition indicate the percent of times that the target-absent category was predicted, when steering towards the target absent category (false alarm). Error bars indicate the 95% bootstrapped confidence interval across composite images. Horizontal, dashed gray lines indicate the accuracy of classifying the individual target images alone, computed for the LRM3 model for its default modulated pass.

## A.6 Steering modulates face-tuned and scene-tuned units

Classic studies in cognitive neuroscience have shown that category- and object-based attention dramatically alter neural responses in the ventral stream, even within category-selective brain regions. For instance, in the human visual cortex, there are face-selective and scene-selective brain regions: e.g., the fusiform-face area (FFA) responds more to faces than scenes or any other category of object [71], whereas the parahippocampal place area (PPA) responds more to scenes than to faces or other categories of object [72]. When presented with overlapping faces and scenes, the activity of these

regions varies dramatically based on the attended category [37, 73], demonstrating that top-down guidance signals have a strong influence on processing within the ventral stream (Fig. 6A).

We tested whether the LRM3 model would show similar effects of category-based attention within analogues of the human-FFA and human-PPA. To do so, we employed a method we introduced in prior work [74] to identify both face-selective and scene-selective units within the model using the same stimuli and statistical corrections used in the fMRI literature. Specifically, we used a "category localizer" image dataset composed of 400 images, 80 per category for 5 categories (Faces, Bodies, Objects, Scenes, Words) known to selectively activate different regions of the human visual cortex. We passed each image through the network and stored the activations separately for each layer. To determine whether a unit (channel, x, y) responds selectively to faces, we performed a t-test on activation scores for 80 face images vs. activations to 80 images from another category (separately for each other category one-by-one). The p-values were corrected using a false-discovery-rate (FDR) of .05 to adjust for multiple comparisons. A unit was considered face-selective if its activation was significantly greater for faces than each of the other categories in their head-to-head t-tests. The same procedure was used to identify scene-selective units, identifying units that showed higher activation for scenes than all other categories.

For steering signals, we computed the average model output for all face images (face-prototype) and all scene images (scene-prototype). To determine the effect of steering on face-units and scene-units, we presented overlapping face and scene pairs (each face image was paired with a scene image, and they were blended 50/50: composite1 = .5 * face1 + .5 * scene1). We measured the activation of face-units and scene-units when the network was steered towards the face-prototype, or the scene-prototype, with an alpha of 3.0. As can be seen in Fig. 6B, the activation of model face-units and scene-units is strongly modulated by the steering target, with face-units showing stronger activation when steering towards faces, and scene-units showing stronger activation when steering towards scenes. Thus, LRM models with cognitive steering are able to qualitatively capture this well-known signature of category-based attention in the human brain.

### A.7    Steering in LRM-additive models

Initial analyses were conducted to explore the impact of LRM pathways with *additive* modulatory influence on destination layers. We hypothesized that during cognitive steering, models with additive feedback would have more 'hallucinations' (or false alarms) to the steered category, even if it was not present in the image.

Table 4 focuses on a subset of the results, considering prototype steering, with overlaid images. The LRM3-multiplicative model showed peak performance at $\alpha=3$; the LRM3-additive model showed higher hit rates, but also higher false alarm rates. Further, when steering the LRM-additive models, the false alarm rate climbs with increasing attentional gain ($\alpha=5$), but that does not happen with the LRM-multiplicative models.

Considering the signal detection measure A' (sensitivity of hits relative to false alarms [75]), the LRM3-multiplicative model has an A' that is high and fairly stable across the alpha-gain range. Meanwhile, the LRM3-additive model has an A with a steeper decline of sensitivity with increasing attention modulation. Qualitatively similar results were obtained with the side-by-side composites. Thus, these LRM-additive models help clarify the role of the multiplicative feedback mechanisms in reducing hallucinations during high attentional gain states.

|  | LRM-*multiplicative* | | LRM-*additive* | |
|---|---|---|---|---|
|  | hit rate | % false alarm | % hit | % false alarm |
| LRM3, $\alpha=3$ | 53.3% | 5.8% | 62.3% | **21.2%** |
| LRM3, $\alpha=5$ | 35.0% | **3.2%** | 67.0% | **38.9%** |

Table 4: Comparison between LRM3 models with either multiplicative or additive feedback modulation. These results show the impact of cognitive steering, on the overlaid image composite test.

## A.8 Steering with CORnet

CORnet [40] is a recurrent neural network, with similar consecutive-forward-pass dynamics as LRM models. That is, the image is processed serially through the hierchical layers in the first time step. In CORnet, the activations from each layer are carried forward in time, and then add to the same layer's activation on the next time step (as opposed to earlier layers, as in our LRM models). Pre-trained CORnet models perform 5 forward passes by default.

To outfit this model with steering capacity, we replaced the previous time point's activation with a steering signal. We used a steering signal at every layer at every time point. For instance-level steering, the steering inputs consisted of the activations of the exact image, at every layer and every time point–i.e. perfect oracle instance-level steering. For category-prototype steering, the steering inputs consisted the average activation map over all exemplar images of that category (in the ImageNet Training Dataset), computed at each time step and layer. We additionally included a steering modulation strength factor, multiplying the steering signals by a constant ($s = 1, 2, 3, 4$).

Note that our implementation of CORnet steering uses a stored representation at *every layer* of the visual encoding backbone at *every time point*, per category. This steering interface differs dramatically from the LRM networks, which leverages a single 1000-dimensional vector per category. We note that there are likely many ways to outfit recurrent neural networks to inject steering–we opted for these choices because they seemed the most likely to lead to the largest consequences of cognitive steering.

The results are shown in Figure 9, with varying strengths of both instance steering and prototype steering, assessed on composite overlaid images. It is clear that increasing the strength of steering increases correct target recognition but also introduces false alarms when steering to a category that is not present in the image (target-absent conditions).

We used the signal detection measure A' (A-prime) [75] to quantify this effect. A' is a measure of sensitivity for distinguishing between signal (target present) and noise (target absent), and is a non-parametric version of d' (d-prime). A' values range from 0 to 1, with .5 indicating chance, 1 indicating perfect target detection with no false alarms (values less than .5 are uncommon, e.g., 0 indicates 100% false alarms and 0% hits). The A' effect is plotted for LRM3 and CORnet models on the same plot (though note the way the steering strength is implemented differs between these models). With increasingly strong steering, the LRM models maintain their ability to correctly detect target present categories without false alarms, while the same is not true for the CORnet model (as we have implemented steering here).

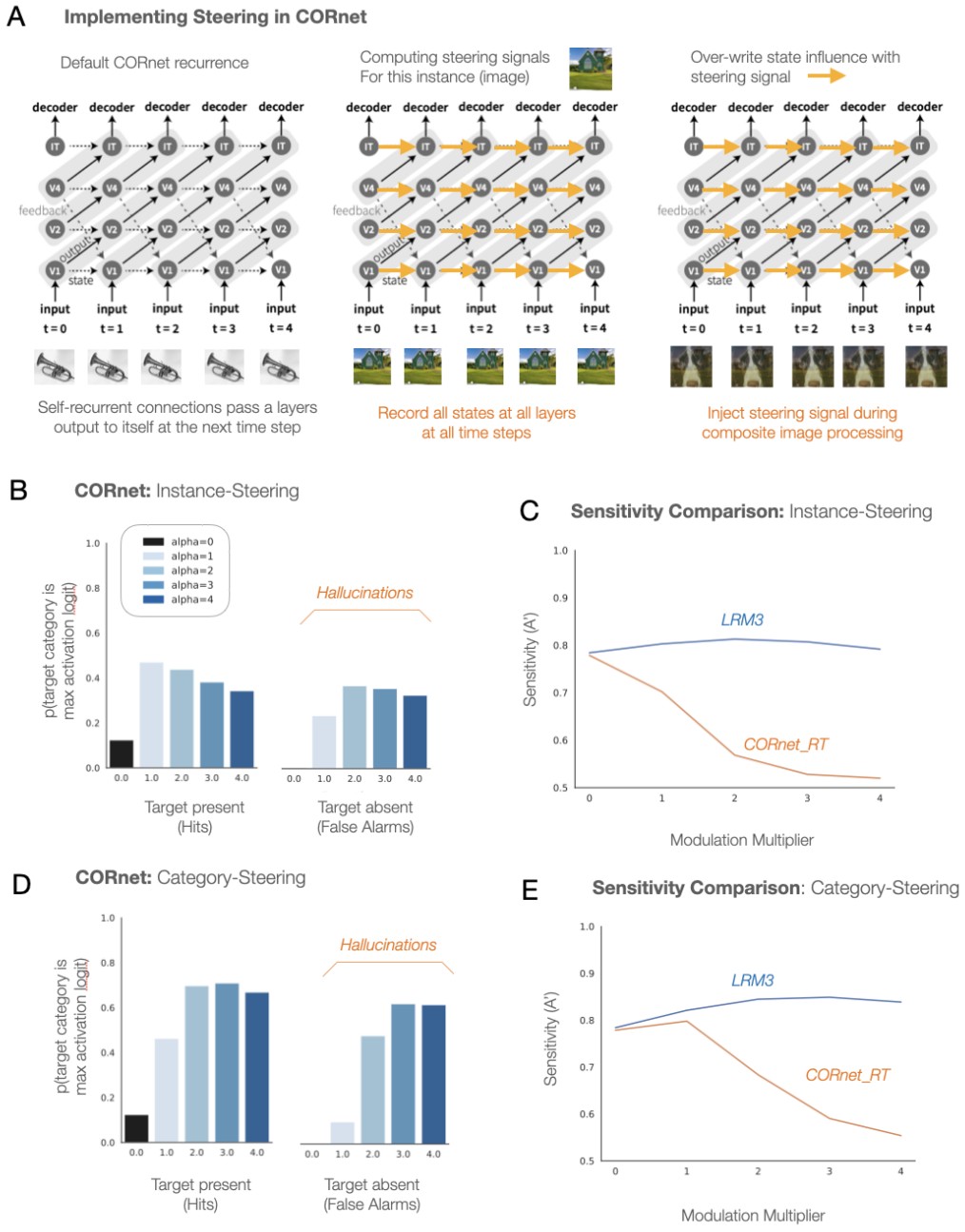

Figure 9: Steering in CORnet. (A) Diagram of the flow of information in CORnet (adapted from Kubilius et al., 2018). Steering signals are calculated for each image (or the average of all images in a category), at each layer and each time point (yellow arrows). During steering, the state is overwritten with these arrows. (B) Instance-steering in CORnet models, on composite overlay images. Probability that a target is the max logit is plotted for steering when the target category is present or absent, for different levels of steering strength. (C) Sensitivity measure, A' (reflecting corrected Hit rate taking into account False Alarms) is plotted for both LRM3 and CORnet_RT models with increasing steering strength. (D,E) Same plots but for category-level steering signals.

