# OpenReview forum: "Cognitive Steering in Deep Neural Networks via Long-Range Modulatory Feedback Connections"
_NeurIPS.cc/2023/Conference — NeurIPS 2023 poster_

### Official Review · Reviewer_kmpt · 2023-07-06

**Soundness:** 4 excellent
**Presentation:** 4 excellent
**Contribution:** 4 excellent
**Rating:** 8
**Confidence:** 4

**Summary:**

This paper explores adding long-range modulatory feedback connections to deep CNNs (specifically AlexNet, evaluated on ImageNet). It explores 2 ways of incorporating the feedback: a default mode and a cognitive steering mode. The results show improvements on ImageNet accuracy, adversarial robustness, and a composite image recognition task.

**Strengths:**

- The paper is high-quality: it's well-written, clear, the visualizations seem to have been very thoughtfully prepared, etc. The motivation for the modulatory connections is well-explained, and the empirical results (ImageNet accuracy, robustness, and cognitive steering effects) are compelling.
- The authors anticipated most of my questions and responded to them in the body of the paper. For example, Section 2.2 has a great description of why certain decisions were made and which options were explored.
- The experiments are explained clearly, and the visualizations really enhance the presentation.
- I think this is a very significant question (how to incorporate long-range feedback into deep neural networks), which has been studied for many years but hasn't quite become mainstream yet. I applaud the authors for thoughtfully probing this question and taking a step toward bringing long-range feedback connections into modern neural networks, which I expect will be a quite impactful addition to NNs when it finally lands.

**Weaknesses:**

- In Section 2.1, I would have liked to see either mathematical equations or pseudocode to remove any ambiguity regarding the implementation of the feedback connections. The descriptions are decent, but I'm having to guess what the exact computations are. The implementation is the essence of the paper, in some sense. It would be nice to make this very precise in the body of the paper.
- It would be good to have a thorough discussion of the costs associated with incorporating these connections (time, memory, etc.). Right now, the paper kind of reads like there's no reason *not* to use them, which probably isn't entirely fair. What am I losing if I incorporate these connections?
- Although the Related Research section is nice, I would like to better understand which 1-3 works are most closely related to this one, with a more detailed description of how this implementation differs from these 1-3 most closely related prior works.

**Questions:**

- What is the precise definition of "modulatory" used here? It seems like one could argue that any feedback connections are ultimately modulatory. What's the exact definition you're using such that other types of feedback connections *aren't* considered modulatory?
- This isn't essential, but I'm curious (and I suspect some readers might be) -- is there something more biologically plausible about this version of feedback connections than some of the prior work?
- How important is the fact that the steering is global vs. local? It might be worth discussing this more.
- In Figure 3, what do you mean by "full branch"?
- In Section 3.2, what label is used?
- At the end of Section 3.1, could you further spell out why relevant features are amplified and irrelevant features are suppressed? It might be helpful to connect this more precisely to the mathematics/implementation, if added to Section 2.1 (as discussed in Weaknesses).
- nit: typo in the second paragraph of Section 3.2 (where --> were)
- In Section 3.3, you're using "target" as the label whereas it's previously used differently in the "source/target" description, right? If so, this dual use might not be optimal. Could you find a way to use 2 different words?
- Is there possibly a better phrase than "target absent category"? It took me a little while to parse this. Do you have any ideas to help clear this up?

**Limitations:**

The paper includes a nice description of the limitations of this work, including limited exploration of different architectures and a lack of mechanistic analysis into why the long-range connections help.

---

> ### Author Rebuttal · Authors · 2023-08-10
>
> # Strengths:
> >*The paper is high-quality: it's well-written, clear, the visualizations seem to have been very thoughtfully prepared, etc. … I think this is a very significant question (how to incorporate long-range feedback into deep neural networks), which has been studied for many years but hasn't quite become mainstream yet.*
>
> Thank you for your positive comments and evaluation of this work’s potential!
>
> # Weaknesses:
> > *In Section 2.1, I would have liked to see either mathematical equations or pseudocode to remove any ambiguity regarding the implementation of the feedback connections.*
>
> We agree this was very unclear (see also R3). We have added a new Section 2.1 clarifying the “Long-range feedback dynamics.” We will also include a mathematical treatment of the modulatory motif in an appendix section, so people do not have to dig into the source code.
>
> > *It would be good to have a thorough discussion of the costs associated with incorporating these connections (time, memory, etc.).*
>
> Generally the main cost of using these models is time: training is slower (~2x), as both the feed-forward and modulatory pass require different computations. Will add this.
>
> > *Although the Related Research section is nice, I would like to better understand which 1-3 works are most closely related to this one, with a more detailed description of how this implementation differs from these 1-3 most closely related prior works.
>
> The two most relevant recent works are (1) Luo, et al. 2021 and (2) Nayebi et al., 2021
>
> (1) Luo, et al. have the same goal: to add category-based attentional steering to models.  However, their implementation does not leverage feedback connections.  Instead, they introduce an attentional weighting layer operation after conv4, where the weights must be learned for each category and each modulation strength (effectively 1000 * 5 different models). Their model also induces extreme false alarms with increasing attentional strength (hence their title about the “costs and benefits of goal-directed attention.”).
>
> (2) Nayebi et al., 2021 is complementary: they actually implemented a huge model search that included architectures with long-range feedback connections from later layers to earlier layers. However, as far as we can tell, the variation in modulatory motifs only included concatenation influences, and they were only focused on the goal of object recognition (not steering).
>
>
> # Questions:
> > *What is the precise definition of "modulatory" used here? It seems like one could argue that any feedback connections are ultimately modulatory.*
>
> True!  We intended to mean the activation of feedback does not induce activation in the absence of feedforward drive (e.g. $x + x*f = 0$ if $x=0$; this is not true with additive recurrent feedback for example, $x+f$). We have clarified this in a new Section 2.1.
>
> > This isn't essential, but I'm curious (and I suspect some readers might be) -- is there something more biologically plausible about this version of feedback connections than some of the prior work?
>
> We’d like to think so, but in truth the detailed connectivity circuits of long-range feedback connections in biological visual system are not known (to our knowledge!). Recent insight into the causal effect of feedback from V4 onto V1, using a very clever optogentics multi-site recording protocol, was recently published in Science this year by Debes & Dragoi, 2023–and we designed our motifs to capture these phenomena. More work to do to compare to biological neural data under different attentional states.
>
> > *How important is the fact that the steering is global vs. local? It might be worth discussing this more.*
>
> Good question! We explicitly designed our feedback globally (channel-to-channel mixing) based on the visual cognition literature, and did not compare to alternative channel x space remixing models; we have added this as a limitation and direction for future work.
>
> > *In Figure 3, what do you mean by "full branch"?*
>
> We meant the depicted branch of the tree that the Ringlet moth is sitting on. We changed the text to be more clear.
>
> > *In Section 3.2, what label is used?*
>
> We are not quite sure what your question is exactly, but we did try to add clarity to this section following comments of Reviewer 1.
>
> > *At the end of Section 3.1, could you further spell out why relevant features are amplified and irrelevant features are suppressed?*
>
> We can only speculate why, and further tests (or theoretical analyses) are required to understand this modulatory “edit” of the feed-forward features. We hypothesize that certain lower-level features are partially informative for some higher-level categories (and not others); thus later evidence of those higher-order categories could be projected back to amplify the earlier stage features (and similarly, suppress the less relevant features).  One possible way to test this would be to analyze the channel-to-channel remixing weights, to explore whether channels linked with positive weights are both more causally involved in recognizing the same categories. We are planning to work on these in future work.
>
> > *In Section 3.3, you're using "target" as the label whereas it's previously used differently in the "source/target" description, right? Could you find a way to use 2 different words?
>
> Thanks for catching this! Feedback pathways now use “source/destination” terminology in Section 2.  And, “targets” now refer exclusively to category labels in Section 3.
>
> —
> (1) Luo, et al (2021). The costs and benefits of goal-directed attention in deep convolutional neural networks. Computational Brain & Behavior.
>
> (2) Nayebi, et al (2022). Recurrent connections in the primate ventral visual stream mediate a trade-off between task performance and network size during core object recognition. Neural Computation.

---

> > ### Comment · Reviewer_kmpt · 2023-08-16
> >
> > Thanks for carefully and thoroughly addressing my questions and concerns!  I am raising my score to an 8.  I stand by my original assessment:
> >
> > "I think this is a very significant question (how to incorporate long-range feedback into deep neural networks), which has been studied for many years but hasn't quite become mainstream yet. I applaud the authors for thoughtfully probing this question and taking a step toward bringing long-range feedback connections into modern neural networks, which I expect will be a quite impactful addition to NNs when it finally lands."
> >
> > I don't think the other reviewers' concerns diminish this.  This paper doesn't *fully* solve the problem of feedback connections, and there are *always* more experiments to run.  However, I don't think that's the goal here.  I think this is a really interesting and important step forward with meaningful conceptual contributions that will help the community move forward.  I hope this work is highlighted at NeurIPS!

---

### Official Review · Reviewer_Bsgy · 2023-07-06

**Soundness:** 2 fair
**Presentation:** 2 fair
**Contribution:** 2 fair
**Rating:** 5
**Confidence:** 3

**Summary:**

The paper presents a novel, brain-inspired modulatory feedback circuitry (long-range modulation, LRM) for regular feedforward DNNs. The multiplicative modulatory pathways can be conditioned on a) higher-level activations (computed in the initial forward pass or subsequent modulatory passes, Sec 2.1, 2.2) to improve the network’s ImageNet accuracy and adversarial robustness (Fig 2) with brain-like dynamics (Fig 3), or on b) steering signals (derived from labels, instance/class activations, CLIP embeddings, etc., Sec 3.3, 3.5) to perform the composite image recognition task (Fig 5, 6). The evaluation is based on AlexNet, with the small-scale composite task created using Imagenette images (side-by-side or overlay, Sec 3.2) following experimental neuroscience protocols.

**Strengths:**

+ [Originality] The paper is sufficiently novel in my opinion, with key architectural features well motivated by experimental psychology & neuroscience evidence and reasonably different from existing RNN & predictive coding based architectures (Sec 5).

**Weaknesses:**

- [Clarity] Although the overall writing is reasonably clear and easy to follow, the ambiguity in technical details renders accurate understanding of the paper impossible without digging into the source code. Examples are as follows.
1) How exactly are the modulatory pathways (Sec 2.2) and subsequent forward passes executed? E.g., in LRM1, is (Conv4 -> Conv1) executed after or concurrently with (Output -> Conv4)? Is the modulatory signal $f$ applied to e.g. $x$ from the initial pass or $x’$ from the first modulatory pass (i.e. $x’ + x*f$, or $x’(1+f)$)?
2) How exactly does the model (likely trained with 224x224 ImageNet images, Sec 2.3) handle both the overlay setting (same image size as training) and the side-by-side setting (2x image size) at the same time?
3) AblationCam (Fig 3), output activation unit (Fig 6), $\sigma\pm$ (Sec 2.1), etc. are undefined.

- [Quality] Empirical evaluation (soundness) is the main issue of the paper. While the proposed composite task and dataset are likely acceptable in psychology & neuroscience papers, it’s unfortunately not really sufficient for conferences like NeurIPS in my opinion. I strongly suggest the authors include additional experiments on more standard (commonly seen) CV datasets, such as [72, 73] or ones from [48-69], and comparisons against (some) existing approaches [48-71] whether they’re mechanistically similar to this work or not.

- [Significance] Although the paper is sufficiently novel, given its non-negligible weaknesses in clarity and quality (soundness), it’s unfortunately hard to conclude that this work is significant (i.e. sufficiently promising). Brain-Score [74] could be a different direction to showcase the paper’s significance.

[70] mixup: Beyond Empirical Risk Minimization, ICLR, 2018.\
[71] CutMix: Regularization Strategy to Train Strong Classifiers with Localizable Features, ICCV, 2019.\
[72] https://paperswithcode.com/dataset/clevr \
[73] https://paperswithcode.com/dataset/multi-dsprites \
[74] https://www.brain-score.org/

**Questions:**

1) Why are the 0th-pass results in Fig 2A and 2C better than the AlexNet baseline? Or, results in L216 better than L176? What does the 0th-pass model have in addition to the baseline?
2) How’s the model’s training & inference speed compared to the baseline? How does the model’s accuracy compare to stronger baselines (either AlexNet with more parameters, or newer networks) running at a similar speed?

**Limitations:**

The authors have sufficiently addressed the paper’s limitations in Sec 4.

---

> ### Author Rebuttal · Authors · 2023-08-09
>
> # Strengths:
> > *[Originality] The paper is sufficiently novel in my opinion, with key architectural features well motivated by experimental psychology & neuroscience evidence and reasonably different from existing RNN & predictive coding based architectures.*
>
> Thanks!
>
> # Weaknesses:
> > *How exactly are the modulatory pathways (Sec 2.2) and subsequent forward passes executed?*
>
> The general flow of information through the network can be best understood as consecutive feed-forward passes. In the initial feed-forward pass ( time $t$), the image input is processed sequentially through the deep neural network as usual. Layers that are the source of feedback to earlier layers will store their output activations, to influence destination layers on the next pass. During the next feed-forward pass ($t+1$)---also referred to in the paper as the "modulated pass"---each stage first computes its typical output, and then checks for any feedback influences ($f_{t}$) coming from the prior pass. The final output of this layer is calculated as $x_{t+1} + x_{t+1} \times f_{t}$.  We have created a new section 2.1 (“Long-range feedback dynamics”), clarifying these important details, and made corresponding changes to Figure 1.
>
> > *How exactly does the model (likely trained with 224x224 ImageNet images, Sec 2.3) handle both the overlay setting (same image size as training) and the side-by-side setting (2x image size) at the same time?*
>
> Our models are able to handle side-by-side images without reducing the scale of the objects presented to the convolutional backbone, by using PyTorch’s built-in adaptive average pooling at the interface between the convolutional and fully-connected layers.  One level more detailed: objects in the side-by-side images (256x512 pixels) are presented at the same physical size (in pixels) to convolutional kernels as when a single image is presented (256x256). The adaptive average pooling operation resizes the output of the final convolutional layer to align with the first fully-connected layer (reshaping from Cx6x12 to Cx6x6). This operation does introduce a slight aspect ratio distortion at the conv5-fc6 interface, but the steering signal is still able to effectively operate over this image to recognize either category.
>
> >*AblationCam (Fig 3), output activation unit (Fig 6), $\sigma\$± (Sec 2.1), etc. are undefined.*
>
> Reference to “AblationCam” added.  Fig 6 caption updated. Attempted further clarification on $\sigma$± definition. Thanks for the careful read.
>
> >*[Quality] Empirical evaluation (soundness) is the main issue of the paper. While the proposed composite task and dataset are likely acceptable in psychology & neuroscience papers, it’s unfortunately not really sufficient for conferences like NeurIPS in my opinion. I strongly suggest the authors include additional experiments on more standard (commonly seen) CV datasets, such as [72, 73] or ones from [48-69], and comparisons against (some) existing approaches [48-71] whether they’re mechanistically similar to this work or not.*
>
> Based on your recommendation, we looked into many models, but as noted in the general rebuttal, most were either not available or trained in outdated coding frameworks (Caffe, LUA-Torch, MATLAB) and/or over simpler datasets (MNIST, Cifar10).  We further worked on implementing steering on a CORnet, even though it was not mechanistically similar to ours–based on your comment! Doing so also helped us clarify the unique contribution of our LRM models.
>
> We see the potential applicability to visual question answering tasks and multi-object array queries as standalone projects, fruitful directions for future work. Our present goal was to get the general infrastructure in place that allows a new approach to these sorts of applications. Which is why we strove for an implementation that is actually agnostic to the task (e.g. any model can have these connections added to it, trained on any objective).  In terms of connecting with CV more, you mentioned Mixup as an augmentation strategy–our overlay composites are essentially mixup with 50/50 blending. We also looked into CutMix, but found many images where the background and inset are no longer identifiable on their own, thus using CutMix this would require further curation. In general, we recognize this is not the most satisfying answer for you, but we think in part the challenge of comparisons here is that goal-directed visual encoding modulation is not (yet?!) an active area of study.
>
> > *[Significance] … Brain-Score [74] could be a different direction to showcase the paper’s significance.*
>
> Indeed, we plan to relate these models to brain data in future work. Until then, we offer that the design and demonstration of networks which can effectively cognitively steer their visual encoding of the input (with minimal hallucination), reflects a novel contribution.
>
> # Questions:
> > *How’s the model’s training & inference speed compared to the baseline? How does the model’s accuracy compare to stronger baselines (either AlexNet with more parameters, or newer networks) running at a similar speed?*
>
> The model takes ~2x as long to run both at training and inference compared to the baseline model. We did not compare to an Alexnet with more parameters, as our goal was not to increase accuracy in the default operation, but to introduce a meaningful interface for top-down cognitive steering.
>
> # Summary:
> Based on your comments, we added much needed technical detail about the multiplicative modulatory operations. While we acknowledge that we have not added brain-score or other human-data measurements, we have attempted to clarify the novel computational contributions (e.g. with the comparison to the CORnet model), and hope that you might even be tempted to slightly increase your score. Thank you.

---

> > ### Comment · Reviewer_Bsgy · 2023-08-17
> > **Re: Rebuttal**
> >
> > Thank you for the response and the CORnet results.
> >
> > The authors stress in the rebuttal that multiplicative feedback is superior to additive feedback based on the CORnet results, which makes me wonder if an additive LRM ($x+f$) really performs worse than the multiplicative version (a fairer comparison). Also, is the multiplicative LRM really hallucination-free by design (L192), or is it just harder to find adversarial examples that cause (strong) hallucination? Finally, what do the hallucinating additive networks (modified CORnet or LRM) attend to (AblationCam) in comparison to the LRM?

---

> > > ### Author Response · Authors · 2023-08-18
> > > **new LRA model leads to more false alarms during steering**
> > >
> > > > *...which makes me wonder if an additive LRM ($x+f$) really performs worse than the multiplicative version (a fairer comparison).*
> > >
> > >
> > > Reasonable question. To answer this, **we created a matched LRA model** which has long-range feedback signal that is *added* to the output. For clarity:
> > >
> > >
> > > LRM: apply multiplicative feedback ($x_{t+1} + x_{t+1} * f_{t}$)
> > >
> > >
> > > ```output = output + output * total_mod```
> > >
> > >
> > > LRA: apply additive feedback ($x_{t+1} + f_{t}$)
> > >
> > >
> > > ```output = output + total_mod```
> > >
> > >
> > >
> > >
> > > Overall, we found that the LRA model trained, and when we implemented prototype-steering this model showed improved detection but also **increased false alarms**.  We have added an Appendix section A.5 with a new results figure detailing these results. To summarize some of them here:
> > > Considering prototype steering, over overlaid images, the LRM3’s peak performance was at alpha=3:
> > > * LRM, alpha=3: hit rate = 53.3%, false alarm = 5.8%
> > > * LRA, alpha=3: hit rate = 62.3%, false alarms = **21.2%**
> > >
> > >
> > > Further, with LRAs the false alarm rate climbs with increasing attentional gain, but that does not happen with the LRMs:
> > > * LRM, alpha=5: hit rate = 35.0%, false alarm = **3.2%**
> > > * LRA, alpha=5: hit rate = 67.0%, false alarms =**38.9%**
> > >
> > >
> > > Considering a measure of A’ (sensitivity of hits relative to false alarms), the LRM model’s A’ is high and fairly stable across the alpha-gain range, while LRA’s show a steeper decline of sensitivity with increasing attention modulation (and CORnets show the steepest decline). Qualitatively similar results were obtained with the side-by-side composites.
> > >
> > >
> > > Thus, the LRA model (and CORnet model) help clarify the role of the multiplicative feedback mechanisms in reducing hallucinations during high attentional gain states. Thanks for your helpful queries. We will create a new section in the manuscript: “ 3.6 Comparisons to additive feedback modulation motifs”, in which we will describe both the LRA and CORnet results with extended plots in the Appendix section A.4 and A.5.
> > >
> > >
> > > > *Also, is the multiplicative LRM really hallucination-free by design (L192), or is it just harder to find adversarial examples that cause (strong) hallucination?*
> > >
> > >
> > > Good catch: any statements of ‘hallucination-free’ are overly strong and we have toned down sentences that imply this throughout the manuscript.
> > >
> > >
> > > It is not impossible to hallucinate in the LRM models–in principle, if the target-absent category is similar enough to the depicted category in the input, you can false alarm by amplifying misleading bits of evidence (e.g. as noted in conversation with Review esAd: if the target is a kit-fox and the image depicts an arctic-fox, category-based steering might result in a false alarm; we haven’t tested the consequence of representational similarity yet).
> > >
> > >
> > > However, certainly our model shows much reduced false alarms, which we attribute to the fact that we are amplifying the input features, rather than injecting the goal directly into the feedforward pass.  So, your statement that ‘it is harder to find adversarial examples that cause strong hallucination”, to us, is kind of a statement of the result: under controlled, identical stimulus conditions, the LRM models hallucinate less than LRA and CORnet models.  Hopefully this result is clearer with the new results section and edits throughout.
> > >
> > >
> > > > *Finally, what do the hallucinating additive networks (modified CORnet or LRM) attend to (AblationCam) in comparison to the LRM?*
> > >
> > >
> > > We actually have not done the AblationCam tests during cognitive steering. (We used it in the manuscript to gain an intuition for how the modulated pass could be better than the feed-forward pass, when there was no steering at all.)  This is a good idea, and given more time we would do this. However, we opted to spend our time on other additional analyses (per Review sqng), as the AblationCam analyses give rather qualitative results on single images and are harder to aggregate, in our experience.
> > >
> > >
> > > Broadly, we offer that the LRA and CORnet additional models (both with additive influences) help to sharpen our empirical contributions about the benefits of the multiplicative effect of feedback (low false alarms even with additional attentional gain). And, we highlight this value with respect to the work of [1] which explicitly noted the problem of steering-induced false-alarms.  We have clarified these points in the manuscript, and hope you find the additional mechanistic clarity worth increasing your score again.
> > >
> > >
> > > [1] Xiaoliang Luo, Brett D Roads, and Bradley C Love. The costs and benefits of goal-directed attention in deep convolutional neural networks. Computational Brain & Behavior, 4(2):213– 230, 2021.

---

> > > > ### Comment · Reviewer_Bsgy · 2023-08-21
> > > > **Re: new results**
> > > >
> > > > Thanks for the new results, I’ve raised my score to 5. I highly recommend the authors to further present some (initial) results on standard CV tasks (e.g. VQA) or Brain-Score in the revision/resubmission to better support the significance of this work.

---

> > > > > ### Author Response · Authors · 2023-08-22
> > > > > **LRM3 models have better BrainScores**
> > > > >
> > > > > > *...some (initial) results on … Brain-Score in the revision/resubmission…*
> > > > >
> > > > >
> > > > > We now have initial results to report for BrainScore. **LRM3 models fit high-level IT cortex dramatically better than PyTorch default Alexnet** (e.g. model rank #145 → #35 across submitted models).  More generally, the LRM3 model with default feedback operation shows a better fit to three of the four visual brain regions, both in its feedforward pass and its modulated pass, compared to the pyTorch default Alexnet model.
> > > > >
> > > > >
> > > > > | Brain Area   | Baseline Alexnet | LRM3 (pass0) | LRM3 (pass1) | ∆ (from baseline) | rank change |
> > > > > | ------------ |:----------------:|:------------:|:------------:|:---:|:------------:|
> > > > > | IT           | r=0.358      | r=0.393  | r=0.400  |  **+0.042**| #145 → **#35** |
> > > > > | V4           | r=0.443     | r=0.454 | r=0.467  | **+0.024** | #153 → **#97** |
> > > > > | V2           | r=0.353     | r=0.341  | r=0.333  | -0.020 | #13 → #48 |
> > > > > | V1           | r=0.507     | r=0.492  | r=0.531  | **+0.024**| #68 → **#32** |
> > > > >
> > > > >
> > > > > Note: the comparison here is to PyTorch’s default Alexnet, rather than our Baseline Alexnet (which was trained with the same hyperparameters as our LRMs). We are awaiting that model’s performance but the website crashed and the brainscore people are working on it now.

---

### Official Review · Reviewer_sqng · 2023-07-07

**Soundness:** 3 good
**Presentation:** 3 good
**Contribution:** 2 fair
**Rating:** 4
**Confidence:** 4

**Summary:**

This work introduces a recurrent modulation module that can be added to visual models so that the top layer can project to and influence the earlier layers. The authors find that the model with the feedback projection layers outperforms the baseline feedforward model in both the categorization performance and adversarial robustness. The model is further analyzed to test whether the feedback modulation can be controlled to meaningfully steer the representations. The authors find that the top-layer steering yields significant performance increase when mixed targets are presenting in the same image in a side-to-side or overlaying fashion.

**Strengths:**

The paper is well-written and easy to read. The significant improvement of the feedback-augmented model compared to the baseline model is also interesting. The steering analysis is done on both side-to-side and overlaying settings.

**Weaknesses:**

The idea of adding feedback modulation from top layers to earlier layers is not new. The authors should more clearly discuss the difference between their work and other models.

The performance and robustness evaluation also needs to be more careful. The feedback connection adds more computation and trainable parameters. But the performance is still compared to the baseline model with less computation and trainable parameters. The authors should compare their model to a larger or deeper architecture.

Although the steering analysis shows that the model can reach higher performance, this analysis is kind of circular as the target signal is explicitly used to generate the modulation signal, which makes the performance improvement unsurprising. This steering property of the new model has its potential, as the proposed visual model is now available to be tested in attention experiments just as how humans can be asked to attend to different parts or features in their input. But more tests and experiments to compare models to humans are needed to illustrate this potential.

**Questions:**

See weakness.

**Limitations:**

Yes.

---

> ### Author Rebuttal · Authors · 2023-08-09
>
> # Strengths:
> > *The paper is well-written and easy to read. The significant improvement of the feedback-augmented model compared to the baseline model is also interesting. The steering analysis is done on both side-to-side and overlaying settings.*
>
> Thanks!
>
> # Weaknesses:
> > *The idea of adding feedback modulation from top layers to earlier layers is not new. For example, the CORnet [1] model has this type of modulation architecture and also finds that adding feedback modulation helps performance and also makes the models more brain-like. The authors should more clearly discuss the difference between their work and this model.*
>
> In fact, CORnet does **not** actually have long-range feedback connections, nor does it have any layer-to-layer interactions over time(!). CORnet recurrence is entirely "local" – it operates only within a layer, with the outputs of a layer serving as subsequent inputs to the same layer. There is no long-range, top-down feedback. In contrast, our long-range-modulatory networks connect deep layers to earlier layers via a cascade of long-range feedback connections (e.g., the output layer to FC6, FC6 to Conv4, Conv4 to Conv1). That being said, we agree that adding feedback modulation is a theme in the literature which we now discuss directly in the introduction. (See also general rebuttal).
>
> >*The performance and robustness evaluation also needs to be more careful. The feedback connection adds more computation and trainable parameters. But the performance is still compared to the baseline model with less computation and trainable parameters. The authors should compare their model to a larger or deeper architecture.*
>
> We agree there are more parameters in the models with long-range feedback connections, and now note this caveat in section 3.1. And, we emphasize that models with more parameters (say, e.g. an Alexnet with more channels per layer) would not support cognitive steering, which is the primary aim of this work.
>
> > *Although the steering analysis shows that the model can reach higher performance, this analysis is kind of circular as the target signal is explicitly used to generate the modulation signal, which makes the performance improvement unsurprising.*
>
> We somewhat agree, but with important clarifications. First, not all steering signals were equally effective–e.g. we thought the 1-hot steering would be the strongest (given that was the objective of the model during training!). However in fact, this was the weakest form of top-down steering, leading to negligible if any benefit (Figure 5.c. left). Thus, injecting category-relevant steering is not guaranteed to lead to improved recognition, as might be expected in a (statistically) circular analysis.  Second, we now more clearly emphasize the key result that the LRM models do not hallucinate a category that is not present in the image. If the steering results were purely circular, such that you always get as output what you inject in as a steering signal, then we would expect to see high false recognition when steering towards a category that is absent. This doesn't occur with our LRM models, but does occur with our CORnet steering tests, which use additive modulation and show dramatic increases in false alarms/hallucinations. In this way, the modulatory motif is also critical. We have clarified these points.
>
> # Summary:
> In following the recommendations of your review, we ourselves learned more details about other recurrent models like CORnet, which we think also might be news for you as well.  Further, based on your comments, we have added a new paragraph and a new analysis attempting to implement steering in a CORnet model, which helps to clarify the benefits of multiplicative modulation, and situate our work more clearly in the literature.  We hope this additional work will entice you to increase your score, even if only a little ; ). Thank you!

---

> > ### Comment · Reviewer_sqng · 2023-08-17
> >
> > Thanks for the detailed response. I was misled by the figures in the CORNet paper and this network indeed does not contain long-range feedback. I will increase my current score to 4 for now.
> >
> > The authors mentioned that the main aim of the paper is to show that the new model supports cognitive steering. I feel that this main aim still needs more evidence. The added results show that modulation feedback is more robust to false alarms than additive feedback. This shows the potential of this architecture, but not enough as now the main aim of the paper is to show how this architecture gives cognitive steering. Just as I asked in my review, can the authors show some examples comparing models to humans to illustrate this potential? Especially, the authors need to show how this cognitive steering ability gave the possibility of modeling certain human behaviors that earlier architectures lacking this ability fail to model.

---

> > > ### Author Response · Authors · 2023-08-20
> > > **Comparisons to category-based attention in human brain responses**
> > >
> > > > *"...the authors need to show how this cognitive steering ability gives the possibility of modeling certain human behaviors that earlier architectures lacking this ability fail to model."*
> > >
> > >
> > > **We agree**, and we have now added a new results section:
> > >
> > >
> > > > **3.5 Comparisons to category-based attention in human brain responses**
> > > >
> > >
> > > > Humans can hear the sentence "look for a face" and then volitionally direct their visual attention to face-like features; extensive research has shown that these goal-based attentional states actually modulate activity along the entire visual processing hierarchy (e.g. evidenced across methods, including single-unit electrophysiology  [15,61], functional magnetic resonance imaging  [62,63], and magneto/encephalography [16,64]; for review see [65,66,67,68]).
> > > >
> > >
> > > >
> > > >Seminal empirical results measured human visual brain responses while viewing face-house composite images, while instructing participants to attend to faces during some scans and to houses during others [62], see also [69,70]. They showed that regions of the visual system with face-selective tuning were more active with attention to faces, while regions with scene-selective tuning were more active with attention to scenes, even though the composite visual stimulus itself was exactly the same.
> > > >
> > >
> > > >
> > > >We tested for these biological signatures in our LRM models.  Specifically, we first identified model units with face-selective tuning or scene-selective tuning, following the procedures developed in [71], using the same stimulus sets as in the neuroimaging experiments. Next, we computed computed face and house prototype steering vectors, and we created composite face-house stimuli, using the same images. We found that steering towards the face prototype increased the activation of face-tuned neurons across many layers of the deep neural network (particularly the late convolutional stages), with the parallel finding when steering towards scenes in scene-selective units (see Section A.4 for method details and results figure).
> > > >
> > >
> > > >
> > > >Thus, these LRM models can recapitulate this hallmark neural signature of top-down category-based attention. To our knowledge, this is the first mechanistic model, operating over rich naturalistic images, to account for these empirical neural signatures. Further, to our knowledge, current feed-forward and recurrent vision models do not (yet) have mechanisms to support top-down goal-based modulation of early visual processing stages (for the most related earlier work see [10,72]).
> > >
> > >
> > >
> > > **We think that this new analysis helps link LRMs to human signatures, in a way that other models without cognitive steering cannot easily do. We hope you agree, and that you will consider further increasing your score. Thanks!**
> > >
> > >
> > >
> > > *references:*
> > >
> > > [10] Wang et al. Attentional neural network: Feature selection using cognitive feedback. Advances in neural information processing systems, 27, 2014.
> > >
> > > [15] Desimone and Duncan. Neural mechanisms of selective visual attention. Annual review of neuroscience, 18(1):193–222, 1995.
> > >
> > > [16] Störmer et al. Tuning attention to object cate- gories: Spatially global effects of attention to faces in visual processing. Journal of cognitive neuroscience, 31(7):937–947, 2019.
> > >
> > > [61] Martinez-Trujillo and Treue. Feature-based attention increases the selectivity of population responses in primate visual cortex. Current biology, 14(9):744–751, 2004.
> > >
> > > [62] O’Craven, et al. fmri evidence for objects as the units of attentional selection. Nature, 401(6753):584–587, 1999.
> > >
> > > [63] Cohen and Tong. Neural mechanisms of object-based attention. Cerebral Cortex, 25(4):1080–1092, 2015.
> > >
> > > [64] Baldauf and Desimone. Neural mechanisms of object-based attention. Science, 344(6182):424–427, 2014.
> > >
> > > [65] Desimone. Visual attention mediated by biased competition in extrastriate visual cortex. Philosophical Transactions of the Royal Society of London. Series B: Biological Sciences, 353(1373):1245–1255, 1998.
> > >
> > > [66] Maunsell and Treue. Feature-based attention in visual cortex. Trends in neurosciences, 29(6):317–322, 2006.
> > >
> > > [67] Noudoost, et al. Top-down control of visual attention. Current opinion in neurobiology, 20(2):183–190, 2010.
> > >
> > > [68] Peelen and Kastner. Attention in the real world: toward understanding its neural basis. Trends in cognitive sciences, 18(5):242–250, 2014.
> > >
> > > [69] Al-Aidroos et al. Top-down attention switches coupling between low-level and high-level areas of human visual cortex. Proceedings of the National Academy of Sciences, 109(36):14675–14680, 2012.
> > >
> > > [70] Furey et al. Dissociation of face-selective cortical responses by attention. Proceedings of the National Academy of Sciences, 103(4):1065–1070, 2006.
> > >
> > > [71] Prince et al. A contrastive coding account of category selectivity in the ventral visual stream. bioRxiv, pages 2023–08, 2023.
> > >
> > > [72] Stollenga et al. Deep networks with internal selective attention through feedback connections. Advances in neural information processing systems, 27, 2014.

---

> > > > ### Author Response · Authors · 2023-08-22
> > > > **additional comparisons to experimental data - BrainScore**
> > > >
> > > > In addition to the new experiment reported in the previous comment, we have also submitted our models to the Brain-Score benchmark and find that **LRM3 models fit high-level IT cortex dramatically better than PyTorch default Alexnet** (e.g. model rank #145 → #35 across submitted models).  More generally, the LRM3 model with default feedback operation shows a better fit to three of the four visual brain regions, both in its feedforward pass and its modulated pass, compared to the pyTorch default Alexnet model.
> > > >
> > > >
> > > > | Brain Area   | Baseline Alexnet | LRM3 (pass0) | LRM3 (pass1) | ∆ (from baseline) | rank change |
> > > > | ------------ |:----------------:|:------------:|:------------:|:---:|:------------:|
> > > > | IT           | r=0.358      | r=0.393  | r=0.400  |  **+0.042**| #145 → **#35** |
> > > > | V4           | r=0.443     | r=0.454 | r=0.467  | **+0.024** | #153 → **#97** |
> > > > | V2           | r=0.353     | r=0.341  | r=0.333  | -0.020 | #13 → #48 |
> > > > | V1           | r=0.507     | r=0.492  | r=0.531  | **+0.024**| #68 → **#32** |
> > > >
> > > >
> > > >
> > > >
> > > > Thus, LRM models in their default operation (no active steering) are more aligned with neurophysiological data, **and** with steering, they can capture the effects of category-based attention in human neurophysiological responses. We hope that you agree these additions better show the potential of LRM models, and hope you find this merits an increase in score.

---

### Official Review · Reviewer_esAd · 2023-07-07

**Soundness:** 3 good
**Presentation:** 2 fair
**Contribution:** 3 good
**Rating:** 7
**Confidence:** 4

**Summary:**

Authors study ways of incorporating cognitive steering in vision neural network models. They add a top-down feedback mechanism to Alexnet with which they report improved performance. Further, they test other steering mechanisms, including prototypes, language-based signals etc. These tests are over image composite tasks where the approaches show greatly improved performance.

**Strengths:**

The paper is interesting, the experiments and the controls are convincing. Cognitive steering in deep CNNs is novel as far as I know, especially with language signals. Some parts of the paper are well written.



**Weaknesses:**

* The paper lacks benchmarking. There are several methods of incorporating feedback in deep learning models from previous years that weren't tested. Look at [1] for a survey. Although I am sympathetic about this since cognitive steering in itself is interesting but the paper needs to be clear that the contributions are in studying various steering methods/signals and not introducing feedback itself.

* Side-by-side composition is not a straightforward task setting - putting images side by side reduces the scale of the objects and since CNNs are not scale invariant that poses a challenge. So I am not quite convinced it is a good test for steering.



[1] Kreiman, G., & Serre, T. (2020). Beyond the feedforward sweep: feedback computations in the visual cortex. Annals of the New York Academy of Sciences, 1464(1), 222–241. https://doi.org/10.1111/nyas.14320

**Questions:**

Improvements to text and minor corrections:
* Please make it clear in the text what "target absent" control means. Only place it is explained is the Figure 5 caption. I had to spend a lot of time trying to understand that until I stumbled on Fig 5 caption.
* Please consider updating Figure 3 to say "target unit" or "target neuron" or "target logit" instead of "target"
* Line 169: where &rarr; were

Questions:
* Why does LRM models have higher accuracy than alexnet at 0th modulatory pass? They should be same?
* In the "target absent" control - how is the absent target chosen? Is it average of every other (998 other target classes not present in the composite or some random class?).
* How many modulatory passes were they trained for and tested for? Is it the same number of passes in training and testing?

---

> ### Author Rebuttal · Authors · 2023-08-09
>
> # Strengths:
> > *The paper is interesting, the experiments and the controls are convincing. Cognitive steering in deep CNNs is novel as far as I know, especially with language signals. Some parts of the paper are well written.*
>
> Thanks!
>
> # Weaknesses:
> > *The paper lacks benchmarking. There are several methods of incorporating feedback in deep learning models from previous years that weren't tested. … the paper needs to be clear that the contributions are in studying various steering methods/signals and not introducing feedback itself.*
>
> Based on your and other reviewers’ feedback, we have (1) added a new paragraph in the introduction that directly states this, and (2) added a new analysis of CORNet (as per R2). CORnet shows high false-alarms when steering to a category that is not present in the image, unlike LRM models. Details in general rebuttal.
>
> > *Side-by-side composition is not a straightforward task setting - putting images side by side reduces the scale of the objects and since CNNs are not scale invariant that poses a challenge. So I am not quite convinced it is a good test for steering.*
>
> Our models are able to handle side-by-side images without reducing the scale of the objects presented to the convolutional backbone, by using PyTorch’s built-in adaptive average pooling at the interface between the convolutional and fully-connected layers.
>
> One level of explanation deeper: objects in the side-by-side images (256x512 pixels) are presented at the same physical size (in pixels) to convolutional kernels as when a single image is presented (256x256) — we’ve basically “expanded the model’s view of the world at a single scale.” The adaptive average pooling operation resizes the final convolutional layer to align with the first fully-connected layer (reshaping from Cx6x12 to Cx6x6). This operation does introduce a slight aspect ratio distortion at the conv5-fc6 interface, but the steering signal is still able to effectively operate over this image to recognize either category, supporting our empirical claims. We do agree that other models may have rigid constraints in input image size, making this a potentially less general benchmark to employ on other models. To this end, the composite overlay task is a complementary task, as it does not have the same resolution issue. We added a note in section 3.2.
>
> # Questions:
> > *In the "target absent" control - how is the absent target chosen?*
>
> We created a set of 3,910 image triplets, randomly drawing 3 images from 3 distinct imagenette categories (without replacement).  Two of the images were used to form a composite image and the third image was used as a “target-absent” or “distractor” category.  This “target-absent” category is critical to test for steering-induced hallucinations, i.e. whether steering towards a non-present category leads the model to report a category that was not present in the input. We have added this text to Section 3.2.
>
> > *How many modulatory passes were they trained for and tested for? Is it the same number of passes in training and testing?*
>
> Only one modulatory pass was ever trained. When assessing default performance (e.g. recognition and robustness, without steering), we ran the model for one or more modulatory passes at test time (to see if additional passes helped). All steering analyses were done with only one modulatory pass. We clarified these method details in section 3.1.
>
> # Improvements to text and minor corrections:
>
> >  *Please make it clear in the text what "target absent" control means. Only place it is explained is the Figure 5 caption. I had to spend a lot of time trying to understand that until I stumbled on Fig 5 caption.*
>
> We have added clarifying text (as in response to Q1 above). Sorry for the confusion, and thanks for sleuthing it out.
>
> > *Please consider updating Figure 3 to say "target unit" or "target neuron" or "target logit" instead of "target"*
>
> We updated it to say “target unit.”
>
> > *Line 169: where → were*
>
> Nice catch (though this text is now changed, as in Q1)

---

> > ### Comment · Reviewer_esAd · 2023-08-17
> > **Reply to authors**
> >
> > Thank you for the detailed responses and additional experiments!
> >
> > > CORnet shows high false-alarms when steering to a category that is not present in the image, unlike LRM models. Details in general rebuttal.
> >
> > That is interesting and surprising, can you speculate as to why? Looking at the rebuttal Fig A, is there a reason why you did not replace/augment the feedback signal (V4 $\rightarrow$ V1) in CORnet with the steering signal? Is the steering signal still "modulatory" as in LRM?
> >
> > > We created a set of 3,910 image triplets, randomly drawing 3 images from 3 distinct imagenette categories (without replacement). Two of the images were used to form a composite image and the third image was used as a “target-absent” or “distractor” category. This “target-absent” category is critical to test for steering-induced hallucinations, i.e. whether steering towards a non-present category leads the model to report a category that was not present in the input.
> >
> > I think there is room for improvement here. The category of the distractor is important, I think. It is probably easier to "hallucinate" a "red fox" rather than a "comic book" while the true category is "kit fox" (all ImageNet categories). Therefore, a more convincing experiment would be to use each one of the other 998 ImageNet categories as distractors and report the worst case accuracy.
> >
> > > Only one modulatory pass was ever trained. When assessing default performance (e.g. recognition and robustness, without steering), we ran the model for one or more modulatory passes at test time
> >
> > I think this can be improved too (although not really necessary for the current work). Recurrent models are known to be brittle when tested for timesteps different from that during training [1] and therefore you might be leaving some accuracy on the table.
> >
> >
> > [1] Drew Linsley, Alekh Karkada Ashok, Lakshmi Narasimhan Govindarajan, Rex Liu, Thomas Serre. Stable and expressive recurrent vision models. NeurIPS 2020. https://arxiv.org/abs/2005.11362

---

> > > ### Author Response · Authors · 2023-08-17
> > > **CORnet does not have long-range feedback; and a note on target-distractor similarity**
> > >
> > > Thanks for your engagement with our work!
> > >
> > > (1) On CORnet:
> > >
> > > Somewhat misleadingly, the CORnet paper includes a dashed long-range feedback lines from V4→V1, implying they included this connection.  In fact, they do not have this long-range feedback in their model, and the paper notes this in the caption “...currently we do not consider models with skip or feedback connections between areas (grayed-out dashed arrows).”
> > >
> > > To outfit their model with steering capacities, without altering it too much for comparative purposes, we intercepted the default CORnet recurrent signal and added the prototype matrix to it. (These prototypes were computed by passing training images from a category through and storing the average CORnet feedback representation, with a different prototype for each region and each time point).  In this way, we tried to stay true to their feedback motif, and simply biased the feedback signals toward the target category.  We speculate that it is the additive nature of their recurrence that makes it prone to hallucination with strong steering signals–as if you’re injecting the category information into the input, rather than amplifying evidence for it.
> > >
> > >
> > > (2) On the triplets:
> > >
> > > We agree–you’re pointing out the importance of target-distractor similarity, which plays an essential role in theories of attentional guidance (e.g. [1], [2]).  The triplets we designed were aimed at testing the efficacy of the *process* of cognitive steering, without running into *representational bottleneck* issues.  (For example, it doesn’t make sense to test steering over two categories that the model cannot even discriminate. So, if we surveyed all 998 categories, this would confound the representation and process–low performance could be due to a failure to steer or a failure to discriminate). Future work could begin to link representation and processes, e.g. looking at steering capacity as a function of target-distractor similarity.  One intriguing possibility is that cognitive steerability over fine-grained differences might be valuable for gaining visual expertise (enabling finer-grained discriminations), this is something we aim to test going forward.
> > >
> > > (3) On more modulatory passes:
> > >
> > > We could indeed train more passes to see how much we are ‘leaving on the table’--thanks for noting this is future work to do. That being said, our networks aren’t brittle: training on 2 passes led to effective, stable performance beyond 2 modulatory passes (which is another difference between our feedback modulation and additive-local-recurrent networks).
> > >
> > > We hope you consider increasing your score.
> > >
> > > [1] Duncan, J., & Humphreys, G. W. (1989). Visual search and stimulus similarity. Psychological review, 96(3), 433.
> > >
> > > [2] Lleras, A., Wang, Z., Ng, G. J. P., Ballew, K., Xu, J., & Buetti, S. (2020). A target contrast signal theory of parallel processing in goal-directed search. Attention, Perception, & Psychophysics, 82, 394-425.

---

> > > > ### Comment · Reviewer_esAd · 2023-08-17
> > > > **Reply to authors**
> > > >
> > > > Thank you again for the response. I agree, many of my earlier issues are resolved and it does warrant an increase in the score.
> > > >
> > > > > (2) On the triplets: We agree–you’re pointing out the importance of target-distractor similarity, which plays an essential role in theories of attentional guidance (e.g. [1], [2]). The triplets we designed were aimed at testing the efficacy of the process of cognitive steering, without running into representational bottleneck issues. (For example, it doesn’t make sense to test steering over two categories that the model cannot even discriminate. So, if we surveyed all 998 categories, this would confound the representation and process–low performance could be due to a failure to steer or a failure to discriminate). Future work could begin to link representation and processes, e.g. looking at steering capacity as a function of target-distractor similarity. One intriguing possibility is that cognitive steerability over fine-grained differences might be valuable for gaining visual expertise (enabling finer-grained discriminations), this is something we aim to test going forward.
> > > >
> > > > This question is essentially trying to think how you would test this model on images where you actually don't know the category. For a new image, what steering signal would you use? One way to do it would be to use the steering signal of each of the 1000 classes. Would it work? If I am understanding correctly, we don't know that since you have not studied the effect of using similar classes are distractors. Is that right?

---

> > > > > ### Author Response · Authors · 2023-08-17
> > > > > **notes on on goal-directed feedback (cognitive steering) vs. unguided feedback (default modulation)**
> > > > >
> > > > > The brute force search algorithm that you propose (i.e. steer to all 1000 categories, one at a time and check) could be implemented. We are not sure what would happen, actually. For example, if you’re looking for a kit fox and you see an arctic fox, it’s possible the differences are amplified, but it could also be that the similarities are amplified. Speculating based on human literature [1], for tight comparisons, a more discriminative steering vector might be more beneficial (i.e. look for arctic-and-not-kit foxes). There are interesting questions, and they are answerable within this framework.
> > > > >
> > > > > A related point is about the framing of your question. We actually think that a *premise* of cognitive steering is that you have a goal or target in mind. That is, you don’t know what the category is that you’re looking at, but you *know what you are looking for*. As an example, the ongoing cognitive task may next require the agent to look for a key, in which case the steering signal might be a key-prototype, and the visual encoder would amplify any responses for key-like features that are present, enabling better detection (without hallucinating keys everywhere).
> > > > >
> > > > > Of note is that the model works even when there is not active cognitive steering happening (i.e. when we haven’t injected a template and only the default activations feedback). We termed this ‘default modulation’.   In this case, the network seems to naturally refine its visual embedding–e.g. Accuracy is improved on subsequent modulatory passes.  In our paper, we highlighted the “fig/ringlet” example as a case where the system is refining its own representation without any cognitive steering, i.e. without knowing the category ahead of time.
> > > > >
> > > > > [1] Watson, A. B., & Rosenholtz, R. (1997). A Rorschach test for visual classification strategies. Investigative Opthalmology and Visual Science, 38, S1.

---

> > > > > > ### Comment · Reviewer_esAd · 2023-08-17
> > > > > >
> > > > > > Thanks!
> > > > > > Score increased from 5 to 7 as my reservations were addressed.

---

> > > > > > > ### Author Response · Authors · 2023-08-17
> > > > > > > **Thanks!**
> > > > > > >
> > > > > > > thanks for your engagement with us and positive re-evaluation!

---

### Author Rebuttal · Authors · 2023-08-10

# Summary of Strengths:

There was near consensus about the **novelty** of introducing cognitive steering capacities into deep neural network models through long-range modulatory feedback connections (R1,R3,R4; though see R2), with experimental results that are  **convincing**, **interesting**, and **compelling** (R1,R2,R4; though see R3), presented in a way that was generally **well-written** (R1,R2,R3,R4), with model design choices that were **well-motivated by experimental psychology and neuroscience** (R3, R4). R4 further notes the quality of visualizations and highlights the future potential of this work, stating that they “expect that bringing long-range feedback connections into modern neural networks… will be a quite impactful addition to NNs.” (Thanks!)

# Summary of Weaknesses:

*Most reviewers noted that adding feedback connections to deep neural networks is not itself new, and our experiments could more clearly relate to prior work (e.g. R1: “[clarify that] the contributions are in studying various steering methods/signals”; R2: “compare to CORnet”, R3: “include … comparison against other approaches”, R4: “include a more detailed comparison to most closely works.”)*

* **We agree.**   We attempted to collect available models with code and weights that use recurrence/feedback in object recognition tasks for direct comparisons. However, most papers we found did not publish code (e.g. 50, 51, 54, 70, 71, 72), had code without pretrained weights (e.g., 52, 56), had unofficial code with changes to the architecture and/or no pretrained weights (e.g., 48, 70), were designed and tested on simpler datasets (e.g. MNIST variations 70, 50, 72], Fashion-MNIST [56], Cifar10[73, 56]), or were implemented in outdated/less widely-used frameworks (e.g. Caffe[75], Lua/Torch[59], Matlab[49], MXNext[63]).  The work in [74] actually implemented long-range feedback connections (with a concatenation motif, no tests of steering); we contacted them and are now working to port their TensorFlow1 implementation into PyTorch. Thus, **we found there are actually relatively few (if any) available models that we could directly test for cognitive steering capacity.**


* Instead, we focused on **CORnet**, which had an open source implementation in PyTorch with pretrained weights.  CORnet does not actually have long-range feedback connections–instead it has within-layer additive recurrence. But, we tried to implement a form of steering anyway by directly injecting a steering signal into the state representation of the recurrent operation (at every layer, at every time step). This steering was able to boost recognition in overlaid composite tests, but also boosted false recognition when steering towards a category that was not present in the image.  This hallucination behavior is due to the *additive* way the recurrent state information is combined with the input in CORnet, which further helps to highlight a key advantage of our *multiplicative* modulatory feedback motif, which has minimal false alarms. **We report this as a new analysis (Section 3.5, Appendix A.4), and the figure is uploaded along with this response.**


* Finally, we added a new paragraph in the introduction, to clarify earlier in the narrative that recurrence and feedback connections have been integrated into deep neural networks in much prior research, and discuss different feedforward-feedback motifs that have been used (e.g. additive, concatenation). The paragraph drives to a clearer statement of how “our work departs from these prior approaches, introducing long-range modulatory feedback pathways in order to support flexible "cognitive steering" of visual encoding.”

*Two reviewers requested *increased clarity for the details of the modulatory motif* and other technical details (R3, R4).*

* **We agree**  and have added a new Section 2.1 clarifying the long-range feedback dynamics, as well as additional notation and edits to Section 2.2 (modulatory motif). Responses to R3 and R4 have further details.

*Two reviewers asked why LRM models have *higher accuracy on the 0th pass* (R1 & R3).*

* **This is in fact a surprising and systematic result.** In these models, the objective of having both a good feedforward and a good modulated pass leads to *better* feed-forward features than a pure feed-forward objective. We highlight this point (Section 3.1), and speculate this arises due to better tuning alignment between features across hierarchical layers.

*The remaining weaknesses and questions were more idiosyncratic to individual reviewers and we address those more extensively below.  Thanks to the reviewers for their time and effort.*

# Summary
Our work introduces cognitive steering into deep neural networks via long-range modulatory feedback, a novelty acknowledged by reviewers.  While adding feedback/recurrence is not new, our unique approach emphasizes "cognitive steering" of visual encoding (e.g., via memory representations or language instructions). We have benchmarked against a modified "Steerable CORnet", highlighting our model's advantages. We believe our contributions offer original advances for the field and hope for the committee's favorable consideration.


References:

[70] Mnih et al. Recurrent models of visual attention. NeurIPS, 2014.

[71] Alom et al. Inception recurrent convolutional neural network for object recognition. Machine Vision and Applications, 2021.

[72] Wang et al. Attentional neural network: Feature selection using cognitive feedback. Neurips, 2014.

[73] Wang et al. Residual attention network for image classification. CVPR, 2017.

[74] Nayebi et al. Recurrent connections in the primate ventral visual stream mediate a trade-off between task performance and network size during core object recognition. Neural Computation, 2022.

[75] Fu et al. Look closer to see better: Recurrent attention convolutional neural network for fine-grained image recognition. CVPR, 2017.

---

### Decision · Program_Chairs · 2023-09-21

**Decision:**

Accept (poster)

**Comment:**

There was unanimous agreement among reviewers that understanding how to incorporate long-range feedback into deep neural networks is significant. There was initial concerns raised by the reviewers regarding experimental limitations but the rebuttal appears to address most of them. One reviewer maintained a relatively negative assessment because they felt that the paper did not make a sufficiently compelling case to justify how this new architecture allows for modeling behavioral data beyond previous models. However, the AC feels that there is enough support from the other reviewers (with one reviewer describing the paper as a "key stepping stone for the community") to warrant acceptance of the paper. Thus the AC recommends the paper to be accepted.